# An internal thioester in a pathogen surface protein mediates covalent host binding

Miriam Walden[1†], John M Edwards[2†], Aleksandra M Dziewulska[2], Rene Bergmann[3], Gerhard Saalbach[1], Su-Yin Kan[2‡], Ona K Miller[2], Miriam Weckener[2], Rosemary J Jackson[2§], Sally L Shirran[2], Catherine H Botting[2], Gordon J Florence[2], Manfred Rohde[3], Mark J Banfield[1*], Ulrich Schwarz-Linek[2*]

[1]Department of Biological Chemistry, John Innes Centre, Norwich, United Kingdom; [2]Biomedical Sciences Research Complex, University of St Andrews, St Andrews, United Kingdom; [3]Central Facility for Microscopy, Helmholtz Centre for Infection Research, Braunschweig, Germany

*For correspondence: mark. banfield@jic.ac.uk (MJB); us6@st-andrews.ac.uk (US-L)

[†]These authors contributed equally to this work

Present address: [‡]School of Diagnostics and Biomedicine, Faculty of Health Sciences, Universiti Sultan Zainal Abidin, Kuala Terengganu, Malaysia; [§]University of Edinburgh, Centre for Cognitive and Neural Systems, Edinburgh, United Kingdom

Competing interests: The authors declare that no competing interests exist.

**Abstract** To cause disease and persist in a host, pathogenic and commensal microbes must adhere to tissues. Colonization and infection depend on specific molecular interactions at the host-microbe interface that involve microbial surface proteins, or adhesins. To date, adhesins are only known to bind to host receptors non-covalently. Here we show that the streptococcal surface protein SfbI mediates covalent interaction with the host protein fibrinogen using an unusual internal thioester bond as a 'chemical harpoon'. This cross-linking reaction allows bacterial attachment to fibrin and SfbI binding to human cells in a model of inflammation. Thioester-containing domains are unexpectedly prevalent in Gram-positive bacteria, including many clinically relevant pathogens. Our findings support bacterial-encoded covalent binding as a new molecular principle in host-microbe interactions. This represents an as yet unexploited target to treat bacterial infection and may also offer novel opportunities for engineering beneficial interactions.

## Introduction

For commensal and pathogenic bacteria, adhesion to host surfaces is a pre-requisite for colonization and infection, and is mediated by surface-presented adhesins (*Pizarro-Cerdá and Cossart, 2006*). Through specific interactions, these proteins can define host and tissue tropism, providing niche environments and a competitive advantage in the search for nutrients. Bacterial adhesins bind either directly to integral host cell surface components, such as integrins or carbohydrates, or they interact with components of the extracellular matrix resulting in indirect binding to receptors on the host cell surface (*Kline et al., 2009*). Such molecular interactions that define the host-microbe interface are generally non-covalent in nature and frequently involve extensive intermolecular interfaces and multivalent binding. The surprising discovery of internal thioester bonds in the pilus tip adhesin Cpa from the Gram-positive human pathogen *Streptococcus pyogenes* raised the possibility of pathogen-encoded covalent adhesion (*Pointon et al., 2010*; *Linke-Winnebeck et al., 2014*). Internal thioester bonds are formed between the side chains of Cys and Gln residues, most likely self-generated by a favorable environment during protein folding. Internal thioesters have previously only been observed in mammalian complement proteins C3 and C4 (*Law and Dodds, 1997*) and related proteins (*Dodds and Law, 1998*; *Lin et al., 2002*; *Cherry and Silverman, 2006*; *Wong and Dessen, 2014*). Complement thioester proteins are large, multi-domain constructs that upon proteolytic activation undergo a conformational change that exposes the reactive thioester (*Janssen et al., 2006*). This is thought to react with nucleophiles on the surface of pathogens, thus mediating irreversible host-encoded covalent tagging of the pathogens for elimination by the host immune system. However, to the best of our knowledge,

**eLife digest** The human body is home to many trillions of microbes; most are harmless, but some may cause disease. To live inside a host, microbes must first attach to host tissues. This process involves multiple proteins on each microbe's surface, called adhesins, which interact with the molecules that make up these tissues.

Like all proteins, adhesins are long chains of simpler building blocks called amino acids, and each amino acid is connected to the next via a strong 'covalent' bond. Adhesins, however, typically attach bacteria to host molecules through the combined strength of many weak 'non-covalent' interactions.

It was recently discovered that one adhesin from a bacterium called *Streptococcus pyogenes* contains a rare, extra covalent bond—called a thioester—in an unusual location between two of its amino acids. *S. pyogenes* is a common cause of throat infections in humans, and can also cause the life-threatening 'flesh-eating disease'.

Walden, Edwards et al. have now used a range of computational, biochemical, structural biology and cell-based techniques to study other adhesins that have thioester bonds in more detail. Computational searches identified hundreds of bacterial proteins containing similar bonds. These included many from bacteria that infect humans: such as *Streptococcus pneumoniae*, which is the most common cause of pneumonia in adults; and *Clostridium difficile*, which is notorious for causing severe gut infections in hospital patients. Closer examination of the three-dimensional structures of three of these proteins—including one called SfbI from *S. pyogenes*—revealed that each had a clear thioester bond. Biochemical tests of an additional nine of the identified proteins strongly suggested they too contained thioester bonds.

Walden, Edwards et al. then showed that SfbI was able to not only attach to tissues like conventional adhesins, but also chemically react with fibrinogen: a human protein that is essential for blood clotting and commonly found in inflamed tissues and healing wounds. This chemical reaction results in the formation of a covalent bond between SfbI and fibrinogen, which is as stable as the bonds that link the amino acids in a protein chain. Further experiments revealed that SfbI strongly binds to human cells grown in the lab under conditions that mimic tissue inflammation. Finally, Walden, Edwards et al. made a mutant version of SfbI that did not contain a thioester, and found that it could not interact with fibrinogen nor bind to human cells.

Together, these findings suggest that thioesters in bacterial adhesins act like 'chemical harpoons', which microbes can use to irreversibly attach themselves to molecules within their host's tissues. This attachment mechanism has not been seen before in host-microbe interactions, and further research is now needed to explore whether interfering with this process could represent a new way to treat bacterial infections.

definitive evidence that internal thioesters in complement proteins deliver an intermolecular bond with a pathogen target is lacking, and it is not clear if target binding is specific or non-specific. Cpa and complement proteins share no identifiable sequence or structural relationship and therefore appear to be evolutionarily distinct. The thioester domains of Cpa are found in a much less complex structural context and do not require proteolytic activation. The only similarity to complement appears to be that the same strategy, a 'chemical harpoon', has evolved independently in bacteria and complement-related proteins for the purpose of irreversible binding.

Internal thioesters are one of the three unexpected self-generating cross-links between amino acid side chains found in subunits of pili and other adhesins of Gram-positive bacteria (*Schwarz-Linek and Banfield, 2014*), with the others being intramolecular isopeptide bonds (*Kang et al., 2007*; *Kang and Baker, 2011*) and ester bonds (*Kwon et al., 2014*). Unlike intramolecular isopeptide or ester bonds, the role of thioesters does not appear to be protein stabilization (*Walden et al., 2014*). Instead, by analogy to complement, thioesters presented on bacterial surfaces may react with nucleophilic groups on host tissue targets, thus establishing pathogen-encoded covalent adhesion. Indeed, binding of *S. pyogenes* to mammalian cells in vitro is severely impaired when an engineered Cpa variant lacking the thioester is expressed (*Pointon et al., 2010*). While this does not confirm covalent attachment, it supports a role for the reactive bond in bacterial adhesion. Interestingly, a recent study showed covalent bond formation between a Cpa thioester domain and the small molecule nucleophile spermidine, confirming the

accessibility and reactivity of the thioester (*Linke-Winnebeck et al., 2014*). However, the molecular mechanisms of targeting and covalent binding to a relevant host receptor have yet to be discovered.

Here we experimentally demonstrate that thioester domains are prevalent in Gram-positive surface proteins, and that they share a conserved three-dimensional structure despite being divergent in protein sequence. This suggests that thioester domains may target distinct receptors and have a widespread but currently unappreciated role in mediating pathogen adhesion to hosts. Further, we present the identification of host fibrinogen as a covalently bound target of the streptococcal surface protein SfbI. In a thioester-dependent mechanism, SfbI reacts with one specific lysine residue in fibrinogen, forming a very stable intermolecular amide bond. Using a combination of computational, biochemical, structural and cell-based assays we reveal a novel mechanism for host adhesion mediated by a pathogen-encoded covalent interaction.

## Results and discussion

### Identification of diverse, putative thioester-containing proteins in Gram-positive bacteria

The discovery of thioester-containing domains (TEDs) in Cpa prompted us to look for similar domains in other proteins. Using a TED of Cpa (Cpa-TED2) as a template, extensive PSI-BLAST similarity searches (*Altschul et al., 1997*) yielded hundreds of potential hits, exclusively in Gram-positive bacteria. 54 sequences from 40 species were analyzed further (*Figure 1—figure supplement 1*). Putative TEDs appear in a variety of domain architectures (*Figure 1*), but are almost always located at

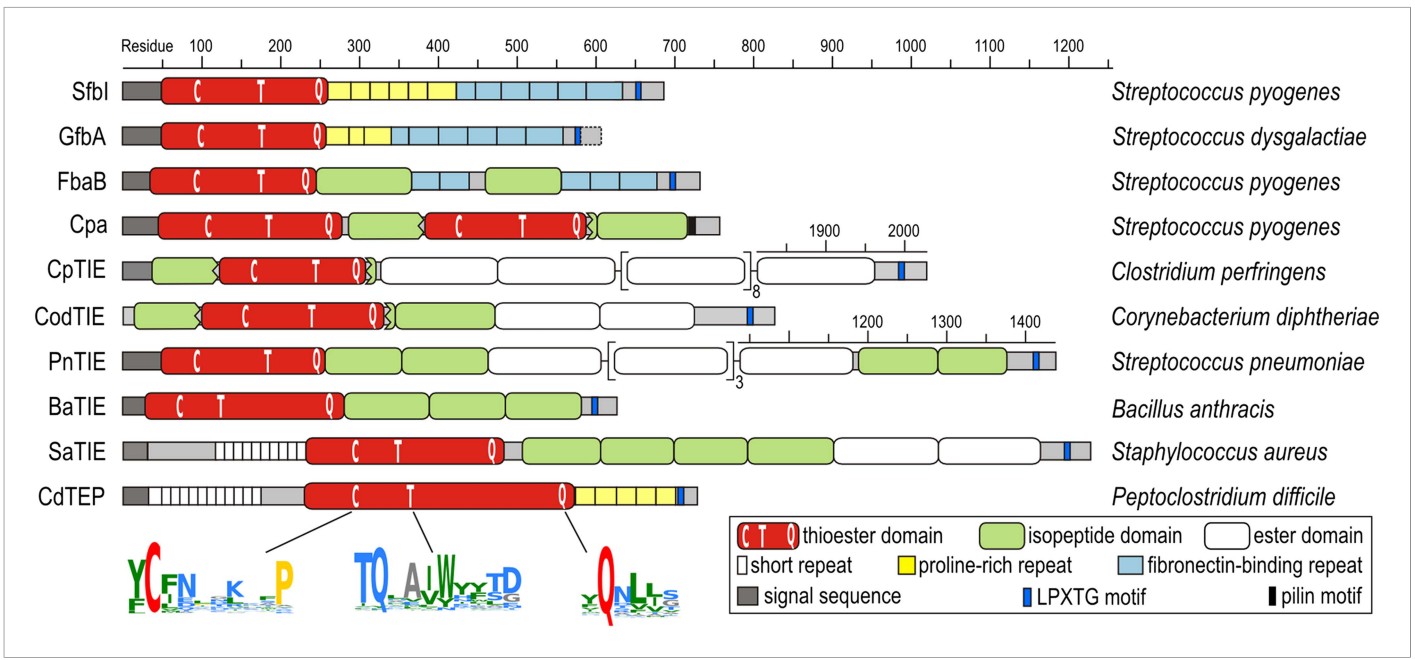

**Figure 1**. Domain architectures of TED-containing proteins of clinically relevant bacteria. C, T and Q indicate positions of conserved motifs. The MEME signatures (*Bailey et al., 2009*) derived from 54 sequences (*Figure 1—figure supplement 1*) are also shown with the thioester-forming residues in red; proline, hydrophobic, polar and small residues in yellow, green, blue and grey, respectively. TIE, thioester, isopeptide, ester domain protein; TEP, thioester domain containing protein.

The following figure supplements are available for figure 1:

**Figure supplement 1**. Sequence alignment of 54 TEDs from 40 species.

**Figure supplement 2**. 10 of the 12 purified TEDs used in this study.

**Figure supplement 3**. Identification of the thioester-forming Gln residues in class-II TEDs.

the N-termini of proteins containing secretion signals and C-terminal LPXTG cell wall-anchoring motifs (*Schneewind and Missiakas, 2014*). Most TED-containing proteins are also predicted to contain intramolecular isopeptide and/or ester domains, usually present in tandem repeat arrays. Like isopeptide domains (sIPDs) in pilus assemblies (*Kang et al., 2007*), these repeats probably act as 'stalks' presenting the TED away from the bacterial surface. We refer to these as TIE (thioester, isopeptide, ester domain) proteins. Other domains commonly associated with putative TEDs are fibronectin-binding repeats and proline-rich regions.

Multiple sequence alignment suggests high diversity among putative TEDs, which can be divided into two classes (class I and class II) based on two indels (*Figure 1—figure supplement 1*). Secondary structure predictions suggest conservation of TED topology, with a central helical region comprising three core helices lying between predicted β-sheet regions. Sequence similarities are mainly limited to three short regions (*Figure 1*). The only fully conserved residues, the thioester-forming Cys and Gln, are found in a [YFL]CΦζ motif (where Φ is any hydrophobic and ζ any hydrophilic residue) and a weak ΦQζΦΦ motif, respectively. Both motifs are consistently predicted to reside in a β-sheet secondary structure context. A TQxxΦWΦxζ α-helical motif, where x is any residue (previously TQxA(I/V)W [*Linke-Winnebeck et al., 2014*]) is conserved in the central region of most, though not all putative TEDs. Mutagenesis of a Cpa-TED demonstrated that neither the Gln nor Trp of this motif is essential for thioester formation (*Linke-Winnebeck et al., 2014*), a notion supported by lack of full conservation of any TQxxΦWΦxζ motif residue (*Figure 1—figure supplement 1*).

## Experimental validation of putative TEDs

To obtain experimental evidence for the apparent abundance and high sequence diversity of TEDs, twelve domains from eight significant human pathogens (*Figure 1*) were recombinantly expressed in *Escherichia coli*. In addition to Cpa-TED2, three allelic variants of the *S. pyogenes* fibronectin-binding protein SfbI (SfbI-A40, SfbI-A346 and SfbI-A20) and its *Streptococcus dysgalactiae* ortholog GfbA were chosen, since their N-terminal domains, now annotated as TEDs, exert an unexplained differential effect on the uptake mechanism of streptococci by mammalian cells (*Rohde et al., 2011*). Other class-I TEDs expressed originate from the fibronectin-binding protein FbaB of *S. pyogenes*, and from TIE proteins of *Clostridium perfringens*, *Corynebacterium diphtheriae* and *Streptococcus pneumoniae*. Finally, three class-II TEDs from *Bacillus anthracis*, vancomycin-resistant *Staphylococcus aureus* isolate VRS11b (*Kos et al., 2012*) and multidrug-resistant *Peptoclostridium difficile* CD630 (formerly *Clostridium difficile*) (*Sebaihia et al., 2006*) were also expressed.

These twelve TEDs were purified to homogeneity (*Figure 1—figure supplement 2*). Each protein displayed an experimental molecular mass, as determined by liquid chromatography–mass spectrometry (LC-MS), ~17 Da lower than predicted (*Table 1*). This is consistent with loss of one molecule of ammonia, as would occur upon internal thioester formation. Seven TEDs were produced as Cys to Ala variants, targeting the Cys of the [YFL]CΦζ motif. For each of these variants, the experimentally determined molecular masses conformed to predicted values (*Table 1*), confirming that the −17 Da differences observed for the native proteins are attributable to internal thioesters. For class-II TEDs (BaTIE, SaTIE and CdTEP) the presence of internal thioesters and the identities of the thioester-forming Gln were established by tryptic-digest LC-MS/MS of the proteins after reacting the thioester bond with the small nucleophile methylamine (*Figure 1—figure supplement 3*). This analysis lends confidence to our definition of domain boundaries of class-II TEDs, which often lack an obvious ΦQζΦΦ motif to define the thioester-forming Gln from sequence alone (*Figure 1—figure supplement 1*).

## TED crystal structures reveal a conserved fold and adaptations that may target different receptors

Crystal structures of SfbI-A40-TED, PnTIE-TED, CpTIE-TED and CpTIE-TED:Cys138Ala were determined (*Figure 2*, *Figure 2—figure supplement 1*; *Table 2*). Despite pairwise sequence identities as low as 12% (*Table 3*), implying no difference from chance, the overall structures of all TEDs solved to date are remarkably similar (*Figure 2—figure supplement 1A*). The three native TED structures determined here show continuous electron density between the Cys and Gln side chains predicted to form the thioesters (SfbI-A40-TED: Cys109-Gln261, PnTIE-TED: Cys94-Gln247, CpTIE-TED: Cys138-Gln267, *Figure 2B*). Consistent with previous data (*Pointon et al., 2010*; *Linke-Winnebeck et al., 2014*),

**Table 1.** Protein intact mass MS analysis

| Protein | Vector | Molecular mass (Da) | | |
| --- | --- | --- | --- | --- |
| | | Calculated | Observed | Δ |
| SfbI-A40-TED | pOPIN-F* | 24,156.9 | 24,139.6 | −17.3 |
| SfbI-A40-TED:Cys109Ala | pOPIN-F* | 24,124.9 | 24,124.7 | −0.2 |
| SfbI-A346-TED | pDEST† | 24,835.8 | 24,818.2 | −17.6 |
| SfbI-A346-TED:Cys103Ala | pOPIN-E‡ | 25,626.6 | 25,626.3 | −0.3 |
| SfbI-A20-TED | pOPIN-E‡ | 25,996.1 | 25,978.5 | −17.6 |
| SfbI-A20-TED:Cys97Ala | pOPIN-E‡ | 25,964.1 | 25,963.6 | −0.5 |
| GfbA-TED | pDEST† | 24,382.4 | 24,364.7 | −17.7 |
| FbaB-TED | pOPIN-F* | 21,683.0 | 21,665.6 | −17.4 |
| FbaB-TED:Cys94Ala | pOPIN-E§ | 22,447.7 | 22,447.5 | −0.2 |
| Cpa-TED2 | pOPIN-F* | 22,377.8 | 22,360.5 | −17.3 |
| CpTIE-TED | pOPIN-F* | 21,261.0 | 21,243.7 | −17.3 |
| CpTIE-TED:Cys138Ala | pOPIN-F* | 21,228.9 | 21,227.8 | −1.1 |
| CodTIE-TED | pOPIN-F* | 25,011.6 | 24,994.3 | −17.3 |
| CodTIE-TED:Cys157Ala | pOPIN-F* | 24,979.5 | 24,979.1 | −0.4 |
| PnTIE-TED | pDEST† | 25,779.0 | 25,761.7 | −17.3 |
| PnTIE-TED:Cys94Ala | pHisTEV† | 25,746.9 | 25,746.5 | −0.4 |
| BaTIE-TED | pHisTEV† | 28,541.1 | 28,523.7 | −17.4 |
| SaTIE-TED | pHisTEV† | 33,167.9 | 33,151.3 | −16.6 |
| CdTEP-TED | pHisTEV# | 46,363.7 | 46,345.7 | −18.0 |

*Non-native residues remaining after 3C cleavage: N-terminal GP.
†Non-native residues remaining after TEV cleavage: N-terminal GAM.
‡Non-native residues remaining: N-terminal M, C-terminal KHHHHHH.
§Non-native residues remaining: C-terminal KHHHHH only. N-terminal M removed.
#Non-native residue remaining after TEV cleavage: N-terminal M.

the thioesters are largely buried at the interface between α-helical and β-barrel subdomains. The thioester Cys backbone carbonyl and amide groups form hydrogen bonds to the Gln, and in some cases Trp, side chains of the TQxxΦWΦxζ motif of the central helix, but from the structures no obvious role of this motif for thioester bond formation is apparent. One notable difference between the TED structures is the position of the loop between the first two strands of the β-barrel, which lies adjacent to the thioester and results in very different protein surfaces around this region (*Figure 2A,C*). Given the proximity to the thioester, we suggest that this loop may be involved in interactions that define thioester target specificity, although this has yet to be tested. Furthermore, the α-helical subdomain shows a larger degree of structural variation than the β-barrel subdomain. The structure of CpTIE-TED: Cys138Ala is essentially identical to the native protein (*Figure 2—figure supplement 1B,C*), confirming that internal thioesters are not structural determinants (*Walden et al., 2014*).

## Thioester-dependent adduct formation of SfbI and FbaB with the Aα subunit of fibrinogen

We next tested the hypothesis that TEDs can target host proteins, forming intermolecular covalent bonds in a thioester-dependent manner. Of our panel of twelve TED proteins, the streptococcal protein SfbI is the most thoroughly characterized. SfbI mediates internalization of bacteria by host cells through binding to the extracellular protein fibronectin (*Schwarz-Linek et al., 2006*). The N-terminal domain of SfbI (revealed here to be a TED) is not strictly required for this process, but defines the uptake mechanism and is a determinant of intracellular bacterial survival (*Rohde et al., 2011*). It has also been reported to interact with fibrinogen (*Katerov et al., 1998*). We investigated if this

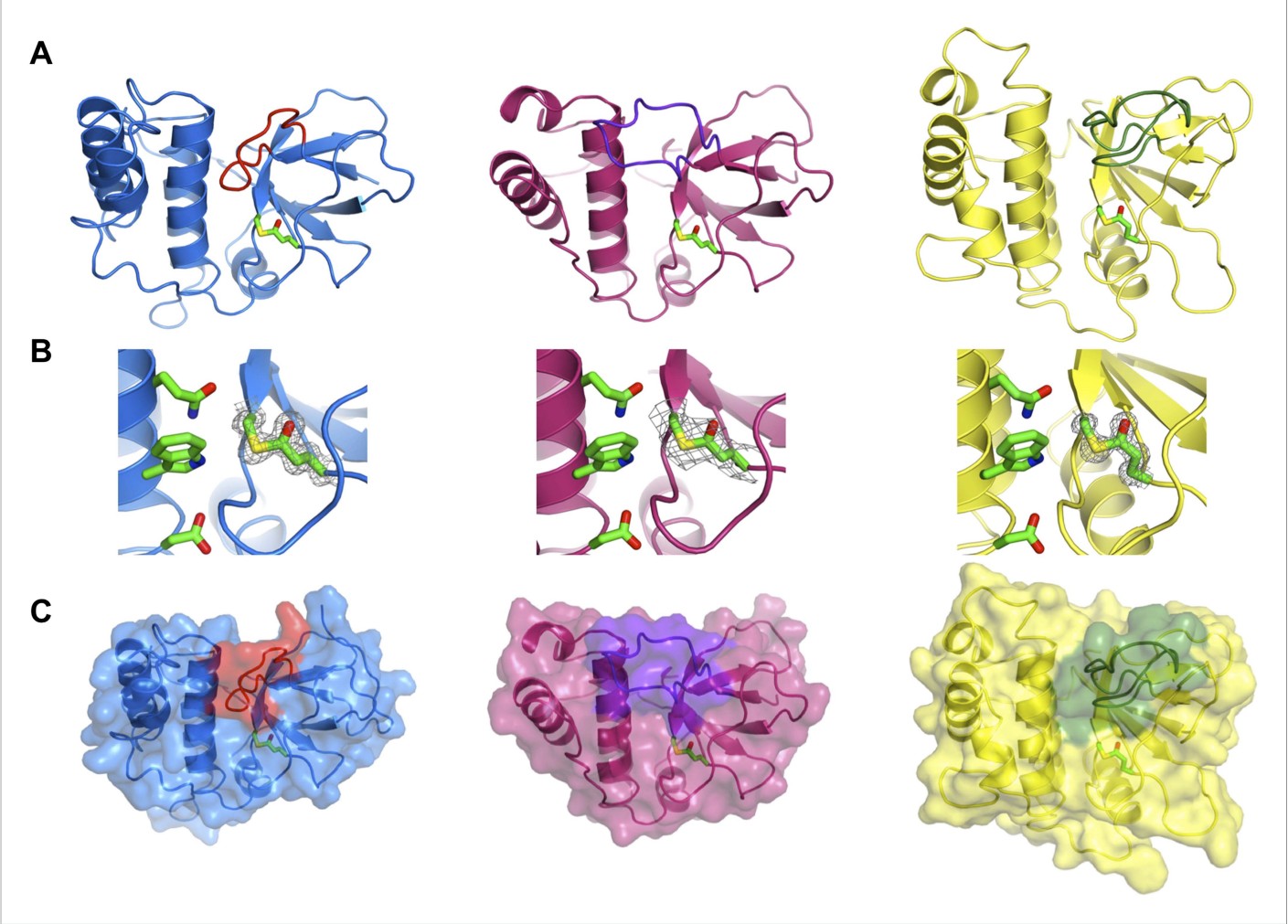

**Figure 2**. Crystal structures of TEDs. (**A**) Cartoon representations of crystal structures of SfbI-A40-TED (blue), CpTIE-TED (burgundy) and PnTIE-TED (yellow) with thioesters shown as sticks. Variable loops adjacent to the thioester are colored red, purple and green, respectively. (**B**) Close-up views of the thioesters. Residues forming the thioesters are shown overlaid with electron density ($2mF_{obs} - DF_{calc}$ contoured at 1.0σ). The 'Q', 'W' and 'ζ' residues of the TQxxΦWΦxζ motif are also shown. (**C**) Surface representations with variable loops colored as in **A**.

The following figure supplement is available for figure 2:

**Figure supplement 1**. Comparison of TED structures.

interaction involves thioester-dependent covalent bond formation by mixing purified TEDs with fibrinogen and analyzing the resulting samples by SDS-PAGE. For all three SfbI-TED variants (SfbI-A40-TED, SfbI-A346-TED, SfbI-A20-TED), and FbaB-TED, a new band is present at a molecular mass consistent with an adduct of each TED with one fibrinogen subunit (*Figure 3A*). Interestingly, a group of protein bands corresponding to the heterogeneous fibrinogen Aα chain (*Mosesson et al., 1972*) show depletion in the adduct samples, which is clearest for SfbI-A20-TED with the conditions used here. This suggested that SfbI-TED and FbaB-TED were covalently binding to the Aα chain of fibrinogen. When fibrinogen is incubated with the Cys/Ala variants of the SfbI-TEDs and FbaB-TED, no adduct bands are observed (*Figure 3B*), suggesting their formation critically depends on the thioesters. Other TEDs did not form a fibrinogen adduct under the conditions of the assay, indicating this activity is not a generic, non-specific property of TEDs.

Tryptic-digest nanoLC-MS[E] of excised adduct gel bands revealed the respective TEDs and the fibrinogen Aα chain as top hits following database searches (*Supplementary file 1*). Peptides from the

**Table 2.** X-ray data collection and refinement statistics

**(A) CpTIE-TED and CpTIE-TED:Cys138Ala**

| | CpTIE-TED | CpTIE-TED:Cys138Ala | |
|---|---|---|---|
| | Native | Native | Iodide |
| Data collection | | | |
| Space group | P1 | P2$_1$2$_1$2 | P2 |
| Cell dimensions | | | |
| a, b, c (Å) | 70.82, 74.36, 82.81 | 98.42, 110.72, 68.77 | 97.91, 110.63, 68.45 |
| α, β, γ (°) | 107.32, 104.32, 98.63 | 90, 90, 90 | 90, 90, 90 |
| Resolution (Å)* | 44.88–2.62 (2.69–2.62) | 44.97–1.60 (1.64–1.60) | 58.21–2.83 (2.90–2.83) |
| R$_{merge}$ | 13.3 (76.4) | 5.4 (74.4) | 19.5 (101.8) |
| I/σI | 7.5 (1.8) | 31.6 (4.3) | 22.7 (4.4) |
| Completeness (%) | | | |
| Overall | 98.0 (97.3) | 100 (99.9) | 98.2 (97.3) |
| Anomalous | | | 98.1 (96.2) |
| Redundancy | | | |
| Overall | 4.7 (4.9) | 18.1 (18.1) | 33.3 (33.0) |
| Anomalous | | | 17.5 (16.8) |
| CC(1/2) (%) | 99.3 (58.9) | 100 (90.9) | 99.8 (86.0) |
| Refinement | | | |
| Resolution (Å) | 44.8–2.62 (2.69–2.62) | 44.97–1.60 (1.64–1.60) | |
| No. reflections | 42,485 (3115) | 94,622 (6864) | |
| R$_{work}$/R$_{free}$ | 19.6/22.6 (35.1/33.7) | 17.7/21.0 (20.5/22.0) | |
| No. atoms | | | |
| Protein | 8260 | 6310 | |
| Ligand/ion/water | 36 | 403 | |
| B-factors | | | |
| Protein | 50.9 | 24.5 | |
| Ligand/ion/water | 37.8 | 28.5 | |
| R.m.s deviations | | | |
| Bond lengths (Å) | 0.014 | 0.013 | |
| Bond angles (°) | 1.60 | 1.57 | |
| MolProbity Score | 1.11 (100th percentile) | 1.13 (99th percentile) | |

**(B) SfbI-A40-TED and PnTIE-TED**

| | SfbI-A40-TED | PnTIE-TED | |
|---|---|---|---|
| | Native/Zinc | Native | Iodide |
| Data collection | | | |
| Space group | I4$_1$ | P4$_1$2$_1$2 | P4$_1$2$_1$2 |
| Cell dimensions | | | |
| a, b, c (Å) | 165.12, 165.12, 42.52 | 59.86, 59.86, 121.70 | 59.37, 59.37, 122.4 |
| α, β, γ (°) | 90, 90, 90 | 90, 90, 90 | 90, 90, 90 |
| Resolution (Å)† | 52.22–1.35 (1.39–1.35) | 42.67–1.30 (1.32–1.30) | 31.22–2.80 (2.95–2.80) |
| R$_{merge}$ | 5.3 (75.4) | 5.4 (30.4) | 13.3 (35.7) |
| I/σI | 18.7 (2.7) | 33.8 (7.4) | 24.9 (13.0) |
| Completeness (%) | | | |

*Table 2. Continued on next page*

*Table 2. Continued*
**(B) SfbI-A40-TED and PnTIE-TED**

| | SfbI-A40-TED | PnTIE-TED | |
| --- | --- | --- | --- |
| | Native/Zinc | Native | Iodide |
| Overall | 99.9 (99.1) | 93.6 (62.7) | 100 (100) |
| Anomalous | 99.6 (97.5) | | 100 (100) |
| Redundancy | | | |
| Overall | 9.2 (9.1) | 23.9 (11.9) | 32.8 (31.9) |
| Anomalous | 4.6 (4.6) | | 18.1 (16.6) |
| CC(1/2) (%) | 99.9 (85.2) | 100 (96.2) | 99.6 (99.2) |
| Refinement | | | |
| Resolution (Å) | 52.22–1.35 (1.39–1.35) | 53.71–1.30 (1.33–1.30) | |
| No. reflections | 119869 (8679) | 48,978 (2374) | |
| $R_{work}/R_{free}$ | 13.1/15.3 (22.6/22.9) | 11.9/15.1 (12.6/15.9) | |
| No. atoms | | | |
| Protein | 3389 | 1848 | |
| Ligand/ion/water | 470 | 252 | |
| B-factors | | | |
| Protein | 23.0 | 12.5 | |
| Ligand/ion/water | 39.4 | 26.9 | |
| R.m.s deviations | | | |
| Bond lengths (Å) | 0.012 | 0.012 | |
| Bond angles (°) | 1.53 | 1.50 | |
| MolProbity Score | 1.06 (99th percentile) | 1.11 (98th percentile) | |

*The highest resolution shell is shown in parenthesis.
†The highest resolution shell is shown in parenthesis.

fibrinogen γ (but not the Bβ) chain were observed at comparably low scores. This can be explained by the presence of a fibrinogen γ dimer, commonly observed in fibrinogen samples, which has a mass similar to the TED-Aα cross-linked protein (*Figure 4—figure supplement 1*). The MS results implicate a site in the fibrinogen Aα chain as the target of SfbI- and FbaB-TEDs. Acetylation of primary amines in fibrinogen abrogates TED/fibrinogen adduct formation (*Figure 3—figure supplement 1*), suggesting Lys side chains as the likely nucleophiles to react with the thioester. This would result in formation of an intermolecular isopeptide bond to the side chain carbonyl group of the TED Gln in the ΦQζΦΦ motif (*Figure 3C*).

## SfbI-TED specifically forms an adduct with fibrinogen Aα in blood plasma

To assess the specificity of SfbI-TED and FbaB-TED binding to fibrinogen, pull-down assays using human blood plasma were performed. TEDs were immobilized using isopeptide domain (IPD) complementation; a strategy that allowed us to use a toolbox approach for many constructs that were used in both pull-down and cell binding (described below) experiments. IPD complementation relies on the spontaneous amide bond formation between a truncated, or split, IPD of the streptococcal FbaB protein that lacks the C-terminal β-strand (sIPD), and a peptide representing this missing strand (*Zakeri et al., 2012*). The peptide, named here isopep-tag (iPT) is used as an expression tag fused as bait to the C-termini of TEDs (TED-iPT) while the sIPD is immobilized on sepharose beads (*Figure 4A*). The advantage of this covalent pull-down strategy is the ability to greatly reduce the amount of non-specific binding. Each of the SfbI-TEDs pulled down fibrinogen from this complex biologically relevant sample by forming an adduct with fibrinogen Aα, although to differing degrees (*Figure 4B*; *Figure 4—figure supplement 1*). For FbaB-TED, little fibrinogen was detectable and the protein

**Table 3.** Pairwise sequence identities of TEDs

| | SfbI-A40 | SfbI-A346 | SfbI-A20 | GfbA | FbaB | Cpa-TED2 | CpTIE | CodTIE | PnTIE | BaTIE | SaTIE | CdTIE | Cpa-TED1 |
|---|---|---|---|---|---|---|---|---|---|---|---|---|---|
| SfbI-A40 | | 54.2 | 54.2 | 52.3 | 27.1 | **25.8** | **27.4** | 24.3 | **12.0** | 7.6 | 7.6 | 7.2 | **50.0** |
| SfbI-A346 | 54.2 | | 49.1 | 64.3 | 22.7 | 21.4 | 31.0 | 23.4 | 9.9 | 6.9 | 6.7 | 8.0 | 46.0 |
| SfbI-A20 | 54.2 | 49.1 | | 56.4 | 26.2 | 23.7 | 27.7 | 21.3 | 8.7 | 8.2 | 5.0 | 7.6 | 52.2 |
| GfbA | 52.3 | 64.3 | 56.4 | | 22.7 | 21.0 | 29.7 | 26.0 | 9.5 | 6.6 | 6.7 | 7.5 | 50.2 |
| FbaB | 27.1 | 22.7 | 26.2 | 22.7 | | 19.1 | 21.9 | 17.7 | 19.5 | 7.8 | 8.0 | 5.5 | 24.8 |
| Cpa-TED2 | **25.8** | 21.4 | 23.7 | 21.0 | 19.1 | | **27.1** | 32.6 | **16.9** | 9.5 | 5.0 | 6.2 | **20.3** |
| CpTIE | **27.4** | 31.0 | 27.7 | 29.7 | 21.9 | **27.1** | | 24.4 | **20.1** | 8.4 | 6.7 | 7.2 | **26.8** |
| CodTIE | 24.3 | 23.4 | 21.3 | 26.0 | 17.7 | 32.6 | 24.4 | | 19.6 | 9.0 | 4.7 | 6.6 | 25.4 |
| PnTIE | **12.0** | 9.9 | 8.7 | 9.5 | 19.5 | **16.9** | **20.1** | 19.6 | | 5.2 | 5.1 | 6.0 | **21.6** |
| BaTIE | 7.6 | 6.9 | 8.2 | 6.6 | 7.8 | 9.5 | 8.4 | 9.0 | 5.2 | | 9.3 | 11.8 | 9.1 |
| SaTIE | 7.6 | 6.7 | 5.0 | 6.7 | 8.0 | 5.0 | 6.7 | 4.7 | 5.1 | 9.3 | | 9.2 | 7.0 |
| CdTIE | 7.2 | 8.0 | 7.6 | 7.5 | 5.5 | 6.2 | 7.2 | 6.6 | 6.0 | 11.8 | 9.2 | | 8.0 |
| Cpa-TED1 | 50.0 | 46.0 | 52.2 | 50.2 | 24.8 | **20.3** | **26.8** | 25.4 | **21.6** | 9.1 | 7.0 | 8.0 | |

Bold values correspond to pairs of TEDs with known structures. Values highlighted in grey indicate that a pairwise alignment was not meaningful; the values given correspond to pairwise identities as calculated from the alignment of 54 TEDs (**Figure 1—figure supplement 1**). Alignments of randomized sequences commonly resulted in pairwise identities of 10–20%. Pairwise alignments were produced with BioEdit using a GONNET similarity matrix.

appeared to bind non-covalently to albumin. All TED Cys/Ala variants failed to pull down fibrinogen from plasma. There is no evidence for covalent binding of TEDs to any other protein in plasma pull-downs. Gel bands present in samples eluted from the beads were analyzed by MS and contained sequences matching fibrinogen, or the TED-sIPD pull-down constructs. For FbaB, the albumin band contained no TED peptides.

## SfbI-TEDs and FbaB-TED specifically target fibrinogen Aα-Lys100

The fibrinogen Aα Lys residue involved in covalent bond formation was identified by searching both low and high trap collision energy nanoLC-MS$^E$ spectra for precursor and fragment ion masses consistent with calculated values for theoretical cross-links. This analysis was carried out for gel bands obtained from experiments using isolated fibrinogen and, for SfbI-A40-TED and SfbI-A20-TED, also for plasma pull-down experiments. Multiple precursor ions and peptide fragments were recovered, all of which were consistent with a single candidate nucleophilic residue, fibrinogen Aα-Lys100 (Lys81 in the mature protein) (**Figure 5**; **Supplementary file 1**), which formed covalent links with the TED Gln residues of the ΦQζΦΦ motifs.

To further support specific targeting of Aα-Lys100 by SfbI-A40-TED, we subjected fibrinogen to acetylation in the absence or presence of SfbI-A40-TED. Acetylation results in specific modification of solvent-accessible Lys ε-amine groups. Following proteolytic digests of bands corresponding to fibrinogen Aα and the SfbI-A40-TED:fibrinogen Aα covalent adduct, we observed near-complete coverage of the Aα sequence (**Figure 5—figure supplement 1**). Of all Aα-Lys residues, only Lys100 was acetylated in the free, but not the bound form of fibrinogen. One other Lys residue, Lys446, was not covered by the analyses of either free or bound fibrinogen. nanoLC-MS$^E$ spectra were scrutinized for potential cross-links between the tryptic peptide containing this Lys residue and TEDs, but no matching precursor peptide masses could be found.

All available evidence supports reaction of SfbI and FbaB TEDs with fibrinogen Aα-Lys100 exclusively. If binding were not specific for a single Lys, we would expect to observe multiple binding of TEDs to different Lys residues on a single fibrinogen molecule. We do not see evidence for such higher-order complexes in SDS-PAGE gels. Further, if only one TED can covalently bind to fibrinogen, but through different Lys residues, this would result in a mixed population of cross-linked species with the same molecular mass, but different acetylation and tryptic digest patterns. There is no evidence to support such events in our exhaustive MS analyses, which only support modification of Aα-Lys100.

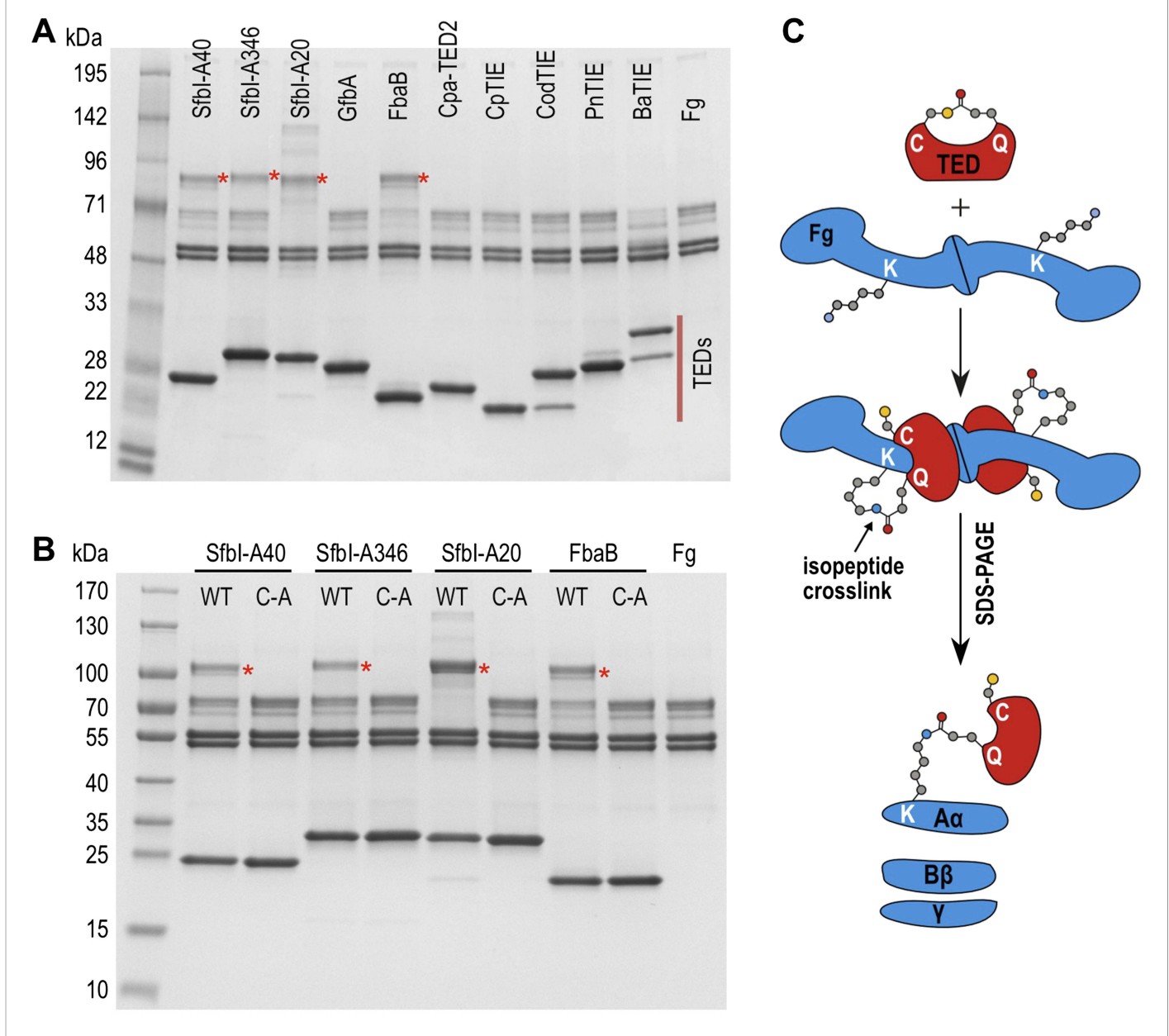

**Figure 3**. SDS-PAGE analysis of TED:fibrinogen (Fg) adduct formation. (**A**) Stable adducts between fibrinogen and TEDs revealed by SDS-PAGE. Red asterisks, adduct bands. (**B**) Cys-to-Ala mutants of TEDs do not result in adduct formation after incubation with Fg. (**C**) Schematic representation of the intermolecular isopeptide bond formed between a Lys residue of fibrinogen Aα and the Gln residue of the TED.

The following figure supplement is available for figure 3:

**Figure supplement 1**. Effect of acetylation of Lys side chains on fibrinogen (Fg):TED adduct formation.

Aα-Lys100 is presented on the surface of the α-helical coiled-coil region of fibrinogen (**Figure 5—figure supplement 2**). It does not participate in the cross-linking of fibrinogen that results in fibrin formation (**Sobel and Gawinowicz, 1996**), but is a plasmin cleavage site (**Kirschbaum and Budzynski, 1990**). This suggests SfbI-TEDs may also bind to fibrin with implications for fibrinolysis. Aα-Lys100 also lies in the vicinity of the integrin-binding RGDF motif of fibrinogen involved in platelet interactions (**Ugarova et al., 1993**).

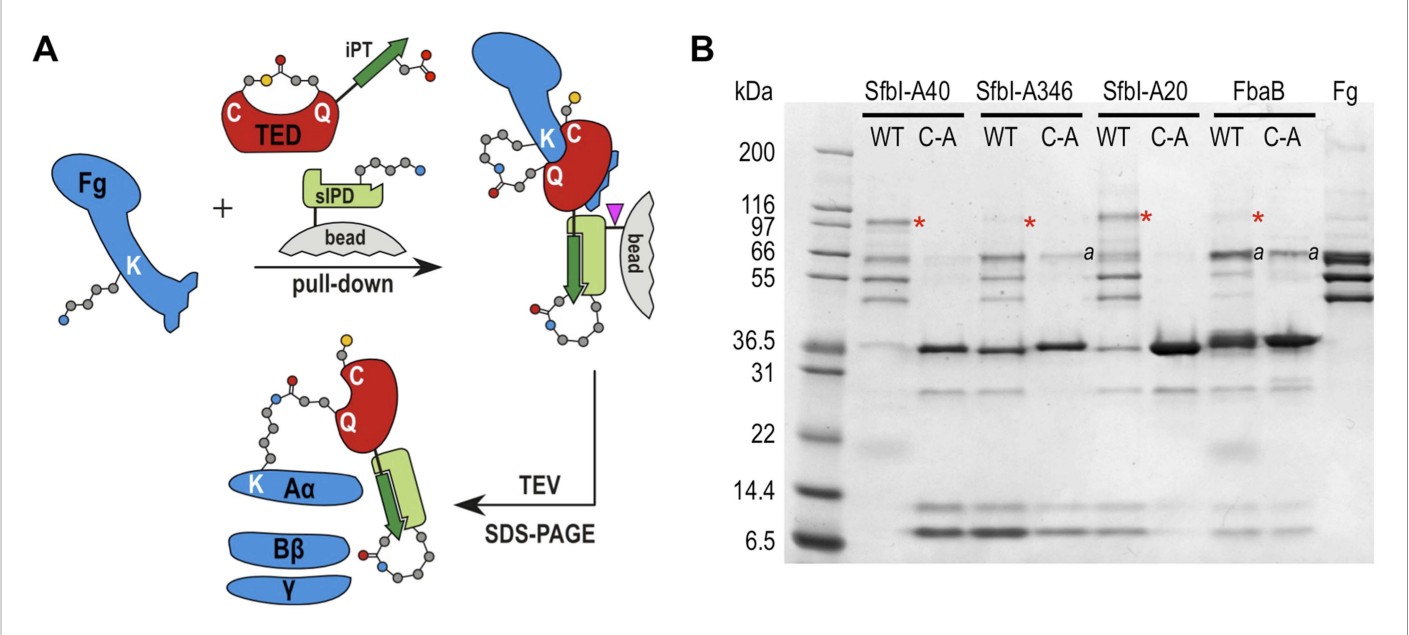

**Figure 4**. TED:fibrinogen (Fg) adduct formation in plasma pull down assays. (**A**) Schematic representation of the isopeptide domain (green) complementation used in the pull-down assays. Pink triangle, tobacco etch virus protease cleavage site. (**B**) SfbI and FbaB-TEDs pull down fibrinogen from blood plasma, forming covalent adducts (red asterisks). '*a*' denotes bands corresponding to human serum albumin. Bands below 31 kDa are breakdown products of tagged TEDs.

The following figure supplement is available for figure 4:

**Figure supplement 1**. TED:fibrinogen (Fg) adduct formation in plasma pull down assays.

## Thioester-dependent bacterial binding to fibrin

To determine a role for thioester-mediated binding to fibrinogen by bacteria, strains of the model Gram-positive bacterium *Lactococcus lactis* expressing either SfbI-A40 or the corresponding SfbI-A40: Cys109Ala variant were produced. Immunogold labeling confirmed the presence of wildtype and variant SfbI at similar levels on the surface of *L. lactis* (*Figure 6—figure supplement 1*). Since fibrinogen Aα-Lys100 is not involved in fibrin formation, it was possible to use a fibrin-based assay for visualization of bacterial binding by electron microscopy. *L. lactis* expressing SfbI-A40, but not bacteria expressing the Cys109Ala variant, show intimate adherence to both single fibrin fibrils (*Figure 6*) and fibrin clots, with the latter reminiscent of biofilms (*Figure 6—figure supplement 2*). Similar results were obtained for *L. lactis* expressing SfbI-A20 and SfbI-A20:Cys97Ala (*Figure 6—figure supplement 2*).

## SfbI-TED binding to human cell surfaces under conditions mimicking inflammation

Fibrinogen is an abundant plasma protein produced mainly by hepatocytes and required for hemostasis. It is however also produced by extrahepatic epithelial cells, and is present in the extracellular matrix (*Pereira et al., 2002*), the provisional matrix in wound healing (*Clark et al., 1982*) and in inflamed tissues (*Lawrence and Simpson-Haidaris, 2004*). Fibrinogen expression is upregulated during inflammatory acute phase response in hepatocytes, but also in epithelial cells by the synergistic action of corticosteroids and interleukin-6 (*Snyers et al., 1990*; *Haidaris, 1997*). This inflammatory response can be induced in cell culture by incubation with dexamethasone and interleukin-6. Under these conditions human A549 lung epithelial cells express and secrete fibrinogen, which remains associated with cell surfaces (*Guadiz et al., 1997*). We investigated if SfbI-TEDs interact with such induced A549 cells. SfbI-TED binding to unfixed, viable, adherent cells was visualized by conjugating TED-iPT constructs via IPD complementation to GFP fused to sIPD (*Zakeri et al., 2012*) (*Figure 7A*). SfbI-A40-TED bound to A549 cells only after they had been pre-incubated with dexamethasone and interleukin-6

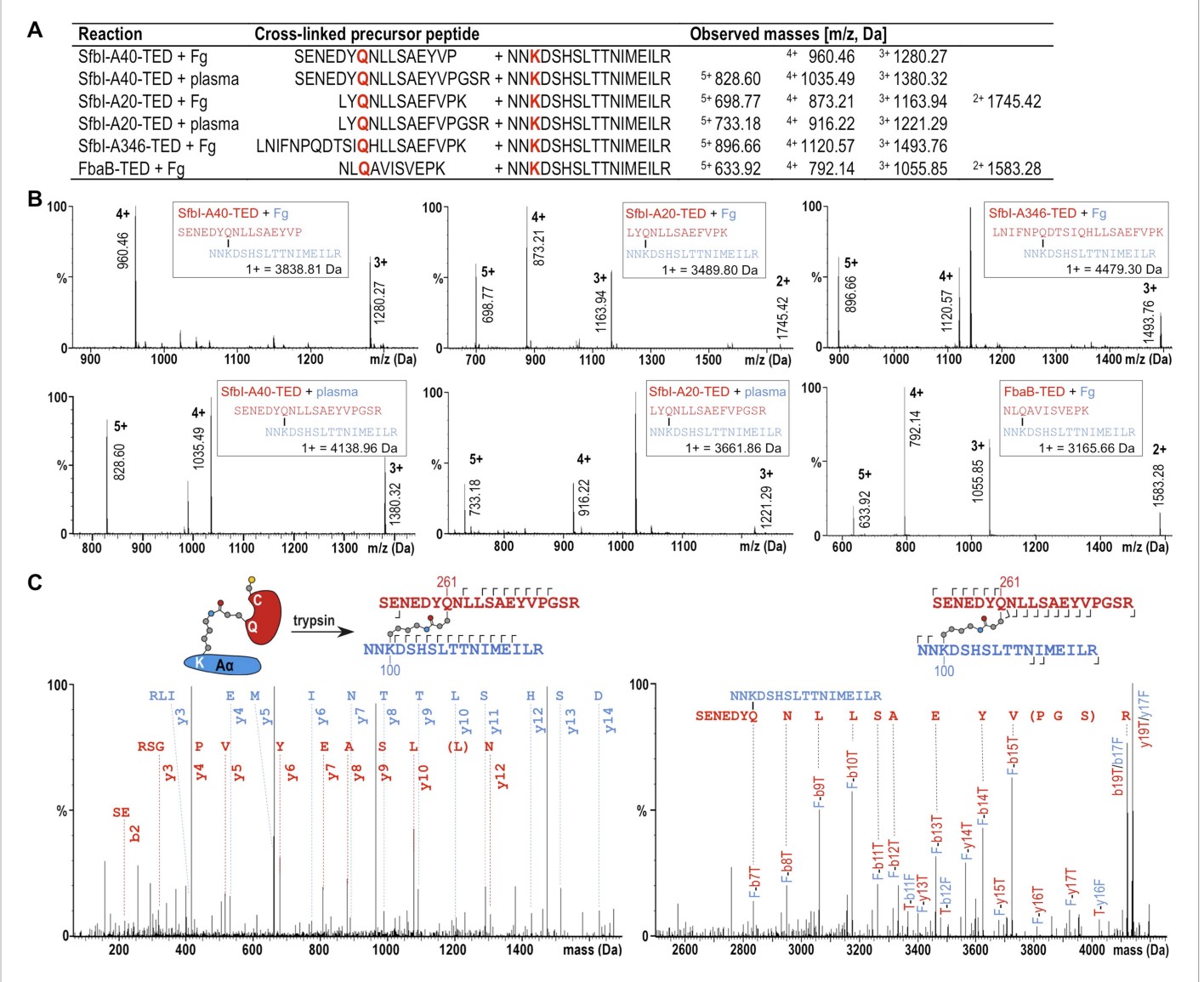

**Figure 5**. Mass spectrometric identification of the SfbI-TED and FbaB-TED target residue in fibrinogen (Fg). (**A**) Precursor ion masses identified in tryptic-digest nanoLC-MS$^E$ of excised adduct gel bands for different samples. (**B**) Several charged states are observed in nanoLC-MS$^E$ of cross-linked precursor peptides obtained by tryptic digestion of adducts formed by TEDs with fibrinogen or in plasma pull-downs. (**C**) Fragmentation nanoLC-MS$^E$ spectra of the cross-linked precursor obtained for SfbI-A40-TED in a plasma pull-down. The low and high mass range spectrum is shown on the left and right, respectively. Fragmentations observed in nanoLC-MS$^E$ are indicated in the schematic drawings above the spectra by hooks, y-series on top of sequences, b-series below sequences. Numbers correspond to positions in the cross-linked fragments; red, SfbI-A40-TED; blue, fibrinogen Aα.

The following figure supplements are available for figure 5:

**Figure supplement 1**. Acetylation of Lys residues in the fibrinogen Aα subunit in absence and presence of SfbI-A40-TED.

**Figure supplement 2**. Structure of fibrinogen with important sites labeled.

(*Figure 7B*). SfbI-A40-TED:Cys109Ala binding to A549 cells was not detectable with or without induction. To obtain direct evidence for fibrinogen binding, Western blot analyses of cell homogenates using an anti-fibrinogen α antibody were performed. These confirmed upregulation of fibrinogen expression following exposure of cells to dexamethasone/interleukin-6, in agreement with published data (*Snyers et al., 1990*; *Haidaris, 1997*). Critically, in homogenates from induced cells incubated with

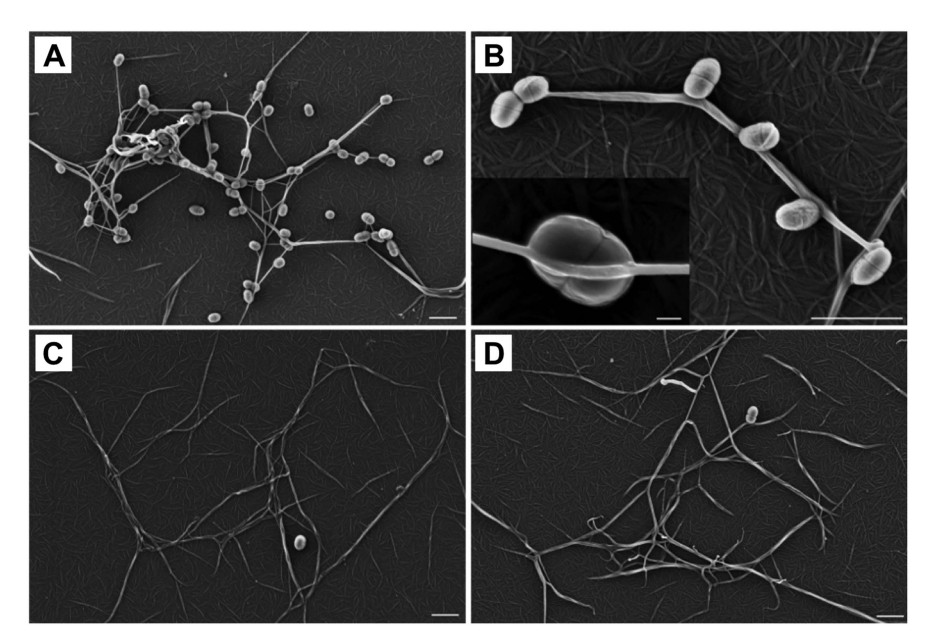

**Figure 6**. Thioester-dependent binding of bacteria to fibrin. (**A**) and (**B**) fibrin binding by *L. lactis* expressing SfbI-A40. (**C**) SfbI-A40:Cys109Ala does not confer fibrin-binding activity to *L. lactis*. (**D**) Control of *L. lactis* transformed with empty pOri23 plasmid. All scale bars 2 μm, except in the insert in panel **B**, 0.2 μm.

The following figure supplements are available for figure 6:

**Figure supplement 1**. Immunogold labeling of SfbI-A40 and SfbI-A40:Cys109Ala on *L. lactis* surfaces.

**Figure supplement 2**. Fibrin clot binding of *L. lactis* expressing SfbI variants.

SfbI-A40-TED, a second band is detected of a molecular mass consistent with an SfbI-A40-TED: fibrinogen Aα adduct (*Figure 7C*). The lack of this band in homogenates from induced cells incubated with SfbI-A40-TED:Cys109Ala confirms formation of this adduct is dependent on the thioester. These data show that SfbI-A40-TED adherence to A549 cells under conditions mimicking inflammation depends on thioester-mediated covalent binding to cell surface-associated fibrinogen.

Pathogenic bacteria disseminated in the blood stream directly or indirectly target fibrinogen and fibrin, resulting in immune evasion, induction of inflammatory response, or platelet activation (*Sun, 2006*; *Rivera et al., 2007*). Two well-studied examples for bacterial fibrinogen-binding proteins involved in invasive infections are clumping factor A of *S. aureus* (*Ganesh et al., 2008*) and the M1 protein of *S. pyogenes* (*Macheboeuf et al., 2011*). In this study we have not investigated the role of SfbI binding to fibrinogen during bacterial infection. However, *S. pyogenes* is known to target epithelial and endothelial cells for adhesion and invasion, and to cause invasive infections involving dissemination in the blood stream. Therefore our experiments using fibrin and epithelial cells represent meaningful biological models to probe the role of thioester-dependent covalent interaction in infections. Perhaps most significantly our data suggest that fibrinogen may have an unappreciated role as a host cell-surface associated adhesion target for bacteria. Potentially SfbI through its fibrinogen-binding activity may confer a tropism for inflamed tissue or the provisional matrix of healing wounds.

To mediate strong attachment, bacterial adhesion complexes typically present extensive, multivalent binding interfaces. Our results show that a wide and diverse range of Gram-positive bacteria have evolved a mechanism for covalent bond formation with specific host factors via a pathogen-encoded reactive internal thioester. Thioester proteins could form a continuous covalent bridge between the bacterial cell envelope and host adhesion targets. This would provide rapid, mechanically very strong attachment that may be particularly advantageous under conditions of shear stress. Lack of similarity in the amino acid sequences of TEDs, together with the observation that only

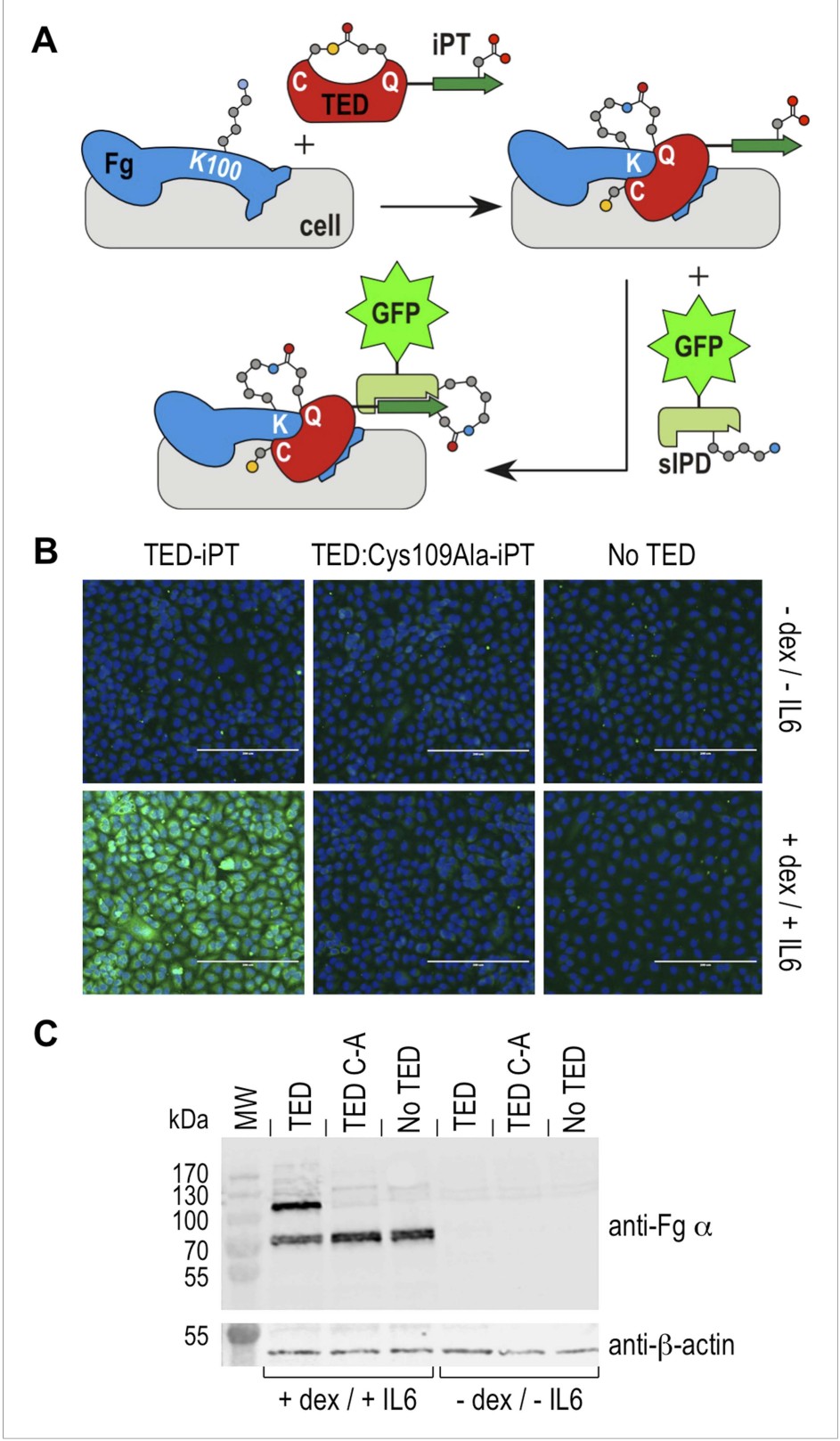

**Figure 7**. Thioester-dependent binding of SfbI-A40-TED to A549 cells. (**A**) Schematic representation of the covalent cell labeling experiment. After incubation with cells, isopep-tag (iPT) labeled TED is detected by GFP fused to sIPD. (**B**) SfbI-A40-TED and SfbI-A40-TED:Cys109Ala binding to A549 cells before and after incubation with interleukin-6

*Figure 7. continued on next page*

*Figure 7. Continued*

(IL6) and dexamethasone (dex). No TED, sIPD-GFP controls. Cell nuclei appear in blue. Scale bars, 200 µm. (**C**) Western blot of whole-cell extracts of induced and non-induced A549 cells, probed with an anti-fibrinogen α antibody. The same membrane was re-probed with an anti-β-actin antibody (shown in separate panel). MW, molecular weight markers; TED, cells incubated with SfbI-A40-TED-iPT; TED C-A, cells incubated with SfbI-A40-TED: Cys109Ala-iPT.

a small subset of those tested here bind fibrinogen, suggests there are undiscovered targets of these proteins. Thioester-dependent covalent adhesion may be a common molecular mechanism in host-microbe interactions, playing a key role in host colonization by both commensal and pathogenic Gram-positive bacteria. As such, TEDs are potentially attractive targets for the design of small molecules to inhibit infection, but they also present an opportunity to engineer beneficial interactions. TEDs hold promise as reactive protein fusion tags for use in tissue engineering, diagnostics or as tools for cell and molecular biology. While we think it is likely that many thioester-containing bacterial proteins will play a role in host-microbe interactions, it is also possible that they function in other processes such as bacteria–bacteria interactions and biofilm formation.

## Materials and methods

### Sequence alignments

The multiple sequence alignment (*Figure 1—figure supplement 1*) was produced with MAFFT/L-INS-i (*Katoh et al., 2002*), refined and formatted using BioEdit (*Hall, 1999*) taking into account structural data and secondary structure predictions by JPred (*Cole et al., 2008*).

### Expression vectors for protein production

Five *E. coli* expression vectors were used to encode TED proteins: pOPIN-F, pOPIN-E, pHisTEV, pDEST and pDEST-iPT. Expression from pOPIN-F results in an N-terminal His$_6$-tag cleavable by 3C protease and from pOPIN-E, a non-cleavable C-terminal His$_6$-tag (*Berrow et al., 2007*). pHisTEV is derived from pET30a (*Liu and Naismith, 2009*) and pDEST from pDEST14. Both incorporate an N-terminal His$_6$-tag cleavable by TEV protease. The pDEST-iPT vector is derived from pDEST with a C-terminal extension consisting of a trypsin cleavage site followed by the iPT sequence AHIVMVDAYK, representing the C-terminal β-strand of the FbaB CnaB2 domain (*Zakeri et al., 2012*).

### Gene cloning for heterologous protein production

DNA encoding the TEDs of SfbI-A40 (Asp63-Pro270), SfbI-A20 (Asp51-Pro262), GfbA (Asp51-Pro256), PnTIE (Gly42-Pro258) and CdTEP (Glu171-Asp578) were amplified from genomic DNA.

The SaTIE ORF is incorrectly annotated in the published genome (*Kos et al., 2012*), appearing fragmented into three ORFs, likely due to short identical sequence repeats. The complete ORF comprises 1223 residues. ORFs I3GYQ5, I3HJ59 and I3HKL2 represent residues 1–179, 172–232, and 281–1223, respectively. Missing residues and the correct length of the repeat region were obtained through cloning and DNA sequencing. The SaTIE-TED construct encompassed Gln254-Ala514 (bases 48,874–49,658 in supercontig 1.1, GenBank AHBV01000001.1).

Constructs of SfbI-A346-TED (Asp51-Pro260) and FbaB (Gly35-Asp365) in pDEST were created by the SPoRT laboratory (University of St Andrews) (*Oke et al., 2010*). The FbaB construct was truncated to FbaB-TED (Gly35-Pro247) by mutation of Trp248 to a stop codon. DNA encoding the TEDs of CpTIE and CpTIE:Cys138Ala (Ser92-Pro277), CodTIE and CodTIE:Cys157Ala (Ser110-Pro327), Cpa-TED2 (Ser390-Pro272) and FbaB:Cys94Ala (Ser56-Pro241) were synthesized and supplied in pUC57 by GenScript (USA). BaTIE-TED (Glu35-Glu288) was amplified from a synthetic gene created by Biomatik (Canada).

Where required, NcoI and BamHI sites were incorporated at the 5′ and 3′ ends respectively. Amplified DNA was inserted into pHisTEV (BaTIE-TED, SaTIE-TED and CdTEP) or pDEST (all others). Cys to Ala point mutants for the SfbI-A40-TED, SfbI-A346-TED, SfbI-A20-TED, FbaB-TED (Gly35-Pro247) and PnTIE-TED were created by site-directed mutagenesis.

A selection of the TEDs were sub-cloned into pOPIN-F or pOPIN-E by In-Fusion (Clontech, France) cloning (*Berrow et al., 2007*). A truncated version of FbaB-TED (Ser56-Pro241) was also sub-cloned

into pOPIN-F and pOPIN-E. The three SfbI-TEDs and FbaB-TED (Gly35-Pro247), and their mutants, were also cloned into pDEST-iPT. For mass spec identification of fibrinogen cross-links, a T231K mutant of FbaB-TED (Ser56-Pro241) in pOPIN-E was synthesised by GenScript (Piscataway, NJ, USA). This allowed generation of a shorter C-terminal Gln234-containing tryptic peptide (NLQAVISVEP), more amenable for mass spec analysis.

The sIPD construct was derived from pDEST-Cna114 (residues 1–113 of the FbaB CnaB2 in pDEST14) (Zakeri et al., 2012). This was shortened to residues 22–107 by deletion of residues 1–21 and mutation of Lys108 to a stop codon. A Cys residue was added N-terminal to the His$_6$-tag, allowing immobilisation on a solid support. The sIPD-GFP construct was generated by insertion of a BamHI site between the His$_6$-tag and sIPD, and cloning of GFP (residues 2–238) between BamHI and NcoI. All primers used for this study are detailed in the Supplementary file 2.

## Protein expression and purification

TED proteins were produced in *E. coli* BL21 (DE3) grown in Luria Broth at 37°C until $A_{600}$ 0.5–0.8. Cultures were induced with 0.5–1 mM isopropyl 1-thio-β-D-galactopyranoside and incubated at 18°C for 16–20 hr.

TED proteins for crystallization, MS and fibrinogen binding assays were expressed from pDEST, pHisTEV, pOPIN-E or pOPIN-F. Cell pellets were re-suspended in 50 mM Tris–HCl or 50 mM bis-tris (pH 7.5 or 6.8 respectively), 500 mM NaCl, 50 mM glycine, 5% (vol/vol) glycerol, 20 mM imidazole, with one EDTA-free protease inhibitor cocktail tablet (Roche) per 40 ml of buffer. Cells were lysed by sonication.

For proteins expressed with a cleavable N-terminal His$_6$-tag (pDEST, pHisTEV, pOPIN-F), clarified lysate was applied to a Ni$^{2+}$-IMAC column (GE Healthcare, UK) and bound proteins were step eluted with buffer (as above) supplemented with 500 mM imidazole. The eluate was loaded onto a Hi-Load 26/60 Superdex 75 gel filtration column (GE Healthcare) pre-equilibrated with 20 mM HEPES (pH 7.5), 150 mM NaCl. Fractions containing TED proteins were incubated with appropriate protease (1:50, wt/wt) at 4°C for 16–20 hr. Imidazole was added to 40 mM and the sample applied to a Ni$^{2+}$-immobilized column. Cleaved TED protein was collected in the flow through, which was concentrated and injected onto the Superdex 75 column pre-equilibrated with either 20 mM HEPES (pH 7.5), 150 mM NaCl, or 20 mM bis-tris (pH 6.0), 150 mM NaCl. Fractions containing purified TED proteins were concentrated to ∼5–10 mg/ml for binding assays and to ∼10–30 mg/ml for crystallization. Protein concentration was determined by $A_{280}$.

For proteins expressed with a non-cleavable C-terminal His$_6$-tag (pOPIN-E), only the initial two-step purification by Ni$^{2+}$-IMAC, and gel filtration chromatography was carried out. Gel filtration buffer was 20 mM bis-tris (pH 6.0), 150 mM NaCl.

TED proteins for cell binding assays and pulldowns from plasma were expressed from pDEST-iPT. Cell pellets were resuspended in phosphate-buffered saline (PBS) (pH 6.0), 10 mM imidazole, with one EDTA-free protease inhibitor cocktail tablet (Roche, UK) per 50 ml of buffer. Cells were lysed by sonication. Clarified lysate was applied to a Ni$^{2+}$-IMAC column and bound proteins eluted with PBS (pH 6.0), 250 mM imidazole. The eluate was dialyzed against PBS (pH 6.0) at 4°C to remove imidazole. The His$_6$-tag was cleaved with His$_6$-tagged TEV protease (1:50, wt/wt) for 16 hr at 4°C. Imidazole was added to 20 mM and the sample applied to a Ni$^{2+}$-immobilized column. Purified TED proteins were collected in the flow through and concentrated to 2–4 mg/ml as determined by $A_{280}$.

The sIPD and sIPD-GFP proteins were expressed from pDEST and purified as above, except that PBS (pH 7.2) buffer was used throughout, and the His$_6$-tag was not removed.

## Intact mass spectrometry analyses

Protein intact masses were determined by LC-MS on a Synapt G2 mass spectrometer coupled to an Acquity UPLC system (Waters, Milford, MA, USA). 50–100 pmol of protein were injected onto an Aeris WIDEPORE 3.6μ C4 column (Phenomenex, UK) and eluted with a 10–90% acetonitrile gradient over 13 min (0.4 ml/min). The spectrometer was controlled by the Masslynx 4.1 software (Waters) and operated in positive MS-TOF and resolution mode with capillary voltage of 2 kV, cone voltage, 40 V. Leu-enkephalin peptide (2 ng/ml, Waters) was infused at 10 μl/min as a lock mass and measured every 30 s. Spectra were generated in Masslynx 4.1 by combining scans and deconvoluted using the MaxEnt1 tool (Waters).

## Identification of Gln residues forming a thioester bond

BaTIE-TED, SaTIE-TED and CdTEP-TED (40 μl at 1 mg/ml in PBS) were incubated with 6 μl 2 M methylamine (in PBS), pH 7.6–8.0 at 25°C for 15 min. The pH was adjusted to 7.0, 4 μl 200 mM

iodoacetamide was added, and the mixture incubated for 60 min at 25°C in darkness. 4 µl 200 mM DTT was added to quench excess IAA.

Protein solutions (5 µl) were dialyzed into 50 mM ammonium bicarbonate, trypsin added and the samples incubated at 37°C overnight. The acidified peptides were separated on an Acclaim PepMap 100 C18 trap and RSLC C18 column (Thermo Fisher Scientific, UK), using a nanoLC Ultra 2D plus loading pump and nanoLC as-2 autosampler (Eskigent, UK). The peptides were eluted with a gradient of increasing acetonitrile. The eluent was sprayed into a TripleTOF 5600 electrospray tandem mass spectrometer (ABSciex, UK) and analyzed in Information Dependent Acquisition mode, performing 250 ms of MS followed by 100 ms MS/MS analyses on the 20 most intense peaks. The MS/MS data file generated in PeakView (ABSciex) was analyzed using the Mascot algorithm (Matrix Science, UK), against an internal database containing the TED sequences as an error-tolerant search, which considers all modifications in Unimod.

## Crystallization, data collection, structure determination and refinement

CpTIE-TED:Cys138Ala crystals were obtained from pOPIN-F-expressed protein, purified at pH 7.5, at ~28 mg/ml (iodide dataset) or ~14 mg/ml (native dataset). Crystals grown in 0.2 M sodium iodide, 0.1 M bis-tris propane (pH 6.5), 20% (wt/vol) PEG 3350, were cryoprotected with mother liquor plus 15% (vol/vol) ethylene glycol. These were used for SAD X-ray data collection and phasing. Crystals grown in 0.2 M potassium thiocyanate, 0.1 M bis-tris propane (pH 6.5), 20% (wt/vol) PEG 3350, were cryoprotected with mother liquor plus 15% (vol/vol) ethylene glycol. These were used for native X-ray data collection.

CpTIE-TED crystals were obtained from pOPIN-F-expressed protein, purified at pH 6.0, at ~26 mg/ml. Crystals grown in 0.2 M tri-potassium citrate, 20% (wt/vol) PEG 3350 were cryoprotected with mother liquor plus 30% (vol/vol) ethylene glycol.

PnTIE-TED crystals were obtained from pDEST-expressed protein, purified at pH 7.5, at ~6.5 mg/ml (iodide dataset) or ~13 mg/ml (native dataset). Crystals grown in 0.1 M sodium acetate (pH 4.6), 25% (wt/vol) PEG 3000, were cryoprotected with mother liquor plus 0.8 M potassium iodide and 25% (vol/vol) glycerol. These were used for SAD data collection and phasing. Crystals grown in the Morpheus Screen (Molecular Dimensions, UK) condition H3, (0.02 M sodium L-glutamate, 0.02 M DL-alanine, 0.02 M glycine, 0.02 m DL-lysine, 0.02 M DL-serine, 0.1 M MES/imidazole [pH 6.5], 10% [wt/vol] PEG 4000, 20% [vol/vol] glycerol), which served as the cryoprotectant also, were used for native data collection.

SfbI-A40-TED crystals were obtained from pOPIN-F-expressed protein, purified at pH 6.0, at ~20 mg/ml. Crystals grown in 0.1 M zinc acetate, 0.1 M HEPES (pH 7.5), 14% (wt/vol) PEG 8000, were cryoprotected with mother liquor plus 20% (vol/vol) glycerol.

Diffraction data were collected at the Diamond Light Source (UK) using beamlines i02 (CpTIE-TED, CpTIE-TED:Cys138Ala, PnTIE-TED) and i04 (SfbI-A40-TED). Data were processed using Xia2 (CpTIE-TED WT, Cys138Ala, SfbI-A40-TED) or iMOSFLM and Aimless in CCP4 (PnTIE-TED) (*Winter, 2010*; *Winn et al., 2011*; *Evans and Murshudov, 2013*). Structures of CpTIE-TED:Cys138Ala, PnTIE-TED and SfbI-A40-TED were solved by SAD phasing using Phenix AutoSol, which also built initial models (*Adams et al., 2010*). For CpTIE-TED:Cys138Ala and PnTIE-TED, the calculated phases were applied and extended to the native data using PARROT (*Cowtan, 2010*) and a model built in the high resolution data using BUCCANEER (*Cowtan, 2006*). For CpTIE-TED, the structure was determined by molecular replacement using PHASER (*McCoy et al., 2007*) with a single chain of the CpTIE-TED:Cys138Ala structure as a search model.

Final models were produced through iterative rounds of refinement using REFMAC5 (*Murshudov et al., 2011*) and manual rebuilding with COOT (*Emsley et al., 2010*). Translation-Liberation-Screw (TLS) and non-crystallographic symmetry restraints were used for CpTIE-TED, whilst only TLS was used for CpTIE-TED:Cys138Ala. TLS groups were based on the α-helical and β-strand subdomains. The β-strand domain comprises Ser100-Gly162 and Pro251-Thr276; the α-helical domain Ser163-Ile250. In Chain C of the CpTIE-TED:Cys138Ala model (PDB:5a0d), it was clear from the electron density that the α-helical domain is somewhat flexible and can be modeled in two distinct positions. For SfbI-A40-TED and PnTIE-TED data, anisotropic B-factor refinement was used. Data collection, phasing and refinement statistics are shown in *Table 2*.

Structure validation was performed using MOLPROBITY (*Chen et al., 2010*) and COOT. Analyses indicated that 98/98/99/97% and 0.19/0.13/0/0% of the residues were in the favored and non-favored

Ramachandran regions for the CpTIE-TED, CpTIE-TED:Cys138Ala, SfbI-A40-TED and PnTIE-TED models respectively.

## Fibrinogen binding assays

Lyophilized human fibrinogen (Sigma, UK) was reconstituted to 2 mg/ml in reaction buffer (20 mM HEPES [pH 7.5], 150 mM NaCl). TED protein purified in 20 mM bis-tris (pH 6.0), 150 mM NaCl was diluted to 10 µM in reaction buffer. Equal volumes of fibrinogen and TED were mixed and left at RT for 60 min. Controls were performed with either fibrinogen or TED replaced with reaction buffer. Samples were analyzed by SDS-PAGE. Bands corresponding to the TED-fibrinogen Aα adduct were excised for analysis.

## Pulldown of TED-fibrinogen complexes from plasma

sIPD protein was immobilized on SulfoLink Coupling Resin (Thermo Fisher Scientific) equilibrated in 50 mM Tris, 5 mM EDTA, pH 8.5 by reaction at 37°C for 45 min. Unreacted iodoacetyl groups were quenched with 50 mM L-cysteine in PBS (pH 7.2) at 37°C for 30 min. After washing (PBS, pH 7.2) the resin was incubated with iPT-TED at 37°C for 30 min. Excess iPT-TED was removed by washing (PBS). The resin was incubated with 1 ml plasma (TCS Biosciences, UK, human plasma mixed pool, citrated) at 37°C for 60 min. For SfbI-A20-TED, prior to this step the plasma was partially depleted of albumin by applying it to a 1 ml HiTrap Blue Sepharose column (GE Healthcare) pre-equilibrated in PBS (pH 7.2). Non-covalently bound proteins were removed by washing (PBS). Protein complexes were cleaved from the resin using TEV protease in PBS (pH 7.2), 1 mM DTT, 0.5 mM EDTA, at 37°C for 60 min. Samples were analyzed by SDS-PAGE. Bands corresponding to the TED-fibrinogen Aα adduct were excised for analysis.

## Covalent TED-fibrinogen binding mapped by mass spectrometry

Excised bands from TED-fibrinogen and TED-plasma binding reactions were digested with trypsin according to standard procedures. Peptides were extracted and analyzed by nanoLC-MS$^E$ on a Synapt G2 mass spectrometer coupled to a nanoAcquity UPLC system. Peptides were trapped using a pre-column (Symmetry C18, 5 µm, 180 µm × 20 mm, Waters), which was switched in-line to an analytical column (BEH C18, 1.7 µm, 75 µm × 250 mm, Waters) for separation. Peptides were eluted with an 8–50% acetonitrile gradient in water/0.1% formic acid at 0.75% per minute (250 nl/min). The column was connected to a 10 µm SilicaTip nanospray emitter (New Objective, Woburn, MA, USA) for infusion into the mass spectrometer. [Glu$^1$]-fibrinopeptide B (1 pmol/µl, Sigma) was infused at 0.5 µl/min as a lock mass and measured every 30 s. The spectrometer was controlled by the Masslynx 4.1 software (Waters) and operated in positive MS$^E$ and sensitivity mode with capillary voltage of 3 kV, cone voltage, 40 V. Scan time was 1 s over the range 50–2000 m/z range. For the low energy scan the trap collision was off, and for the high energy scan the trap collision energy (CE) was ramped from 20–60 V. For de novo identification of proteins, raw files were processed in Protein Lynx Global Server 2.5.2 (Waters), including a search on a *Homo sapiens* protein database to which the TED protein sequences had been added.

Acquisition in MS$^E$ mode uses alternating low and high CE scans and generates 2 separate chromatograms. To detect the TED-fibrinogen Aα cross-linked peptides, extracted ion chromatograms were generated from the high CE trace in Masslynx 4.1. and inspected for characteristic y-ions expected from C-terminal TED-peptide fragments starting with proline. Peptides are known to fragment to the N-terminus of prolines, and the sequences searched for here were Pro (116.071 Da, SfbI-A40-TED-fibrinogen), Pro–Lys (244.166 Da, SfbI-A346-TED- and SfbI-A20-TED-fibrinogen) and Pro-Gly-Ser-Arg (416.226 Da, SfbI-A40-TED- and SfbI-A20-TED-plasma). Detected peaks for those y-ions were aligned with the corresponding low CE traces, which reliably lead to the detection of masses consistent with the cross-linked precursor peptides. In some cases we also searched the low CE trace for masses consistent with all possible combinations of tryptic fibrinogen Aα peptides, cross-linked with the relevant TED peptides, but we could find no evidence for additional combinations. Precursor spectra and high CE fragment spectra were inspected for the presence of characteristic fragment ions of the identified cross-linked peptides.

The lysine-acetylated TED-fibrinogen complex was generated by incubation of 20 µM fibrinogen with a fivefold molar excess of SfbI-A40-TED at 37°C for 1 hr. Sulfo-NHS-acetate (Thermo Fisher Scientific) was then added to a concentration of 10 mM and the reaction mixture incubated at 37°C for 2 hr. The sample components were separated by SDS-PAGE, and the TED-fibrinogen Aα adduct bands excised for analysis by MS. For comparison lysine-acetylated fibrinogen bands were produced by the same protocol, with the omission of the TED reaction step.

Excised gel bands were digested with either trypsin or endoproteinase GluC or, in the case of the TED-fibrinogen adduct, chymotrypsin. The extracted peptides were analyzed by nanoLC-MSMS by the same protocol used for identification of Gln residues involved in thioester bond formation. Peptides with charges +2 to +5 with an ion count over 150 were selected for MSMS, and then excluded for further analysis for 15 s. A rolling CE was applied to fragment the peptides. The data was searched using the Mascot algorithm against an in-house database containing fibrinogen sequences. Carboamidomethyl modification of cysteines was set as a fixed modification, acetylation on lysine and oxidation of methionine were set as variable modifications. MS tolerance was $\pm$ 20 ppm and MSMS tolerance $\pm$ 0.1 Da.

## Lactococcus heterologous protein expression

*L. lactis* was grown at 30°C in M17 medium plus 0.5% glucose (GM17). *S. pyogenes* was grown at 37°C in Todd-Hewitt broth plus 0.5% yeast extract. Where appropriate, antibiotics were added: erythromycin at 3 mg/l for *L. lactis* and 400 mg/l for *E. coli*.

Chromosomal DNA from *S. pyogenes* A40 was prepared as described previously (*Bergmann et al., 2014*) as a template for amplification of the *sfbI-A40* gene. Amplified DNA was inserted into the shuttle vector pOri23 (*Que et al., 2000*) and transformed into *E. coli* XL1-Blue. Site-directed mutagenesis was used to create SfbI-A40:Cys109Ala plasmid DNA. 1 µg of plasmid was used to transform *L. lactis* by electroporation (*Holo and Nes, 1989*).

*L. lactis* constructs SfbI-A40, SfbI-A40:Cys109Ala and pOri23 were grown in GM17 and fixed in the growth medium with 2% formaldehyde. After quenching free aldehydes with 10 mM glycine, samples were incubated with purified polyclonal anti-SfbI IgG-antibodies for 60 min at 37°C, washed and incubated with protein A-gold nanoparticles (15 nm) for 30 min at 37°C. After washing, samples were fixed in 2% glutaraldehyde, absorbed onto butvar-coated 300 mesh grids, washed in distilled water, air-dried and examined in a Zeiss Merlin scanning electron microscope at an acceleration voltage of 10 kV using the high efficiency Everhart-Thornley SE-detector.

## Fibrin binding assays

Fibrin was fibrillized on cover slips by incubating 1 mg plasminogen-depleted human fibrinogen (Calbiochem, Merck Millipore, Billerica, MA, USA) with 10 U of thrombin (from bovine plasma, Sigma) for 12 hr at 37°C in 100 µl PBS. PBS washed *L. lactis* strains, harboring the pOri23, SfbI-A40, SfbI-A40:Cys109Ala plasmids were incubated for 2 hr at 37°C on the fibrin clots. PBS washed cover slips were fixed for scanning electron microscopy with 2% glutaraldehyde and 5% formaldehyde, dehydrated with a graded series of acetone (10, 30, 50, 70, 90, 100%), critical-point dried and sputter-coated with gold-palladium before examination in a Zeiss Merlin field emission scanning electron microscope (Oberkochen, Germany) at an acceleration voltage of 5 kV.

## Lung epithelial cell-binding assays

Human alveolar basal epithelial (A549) cells were propagated in Dulbecco's modified Eagle medium (DMEM; Life Technologies, Thermo Fisher Scientific) supplemented with 10% (vol/vol) fetal bovine serum (FBS, Thermo Fisher Scientific), 100 U/ml penicillin and 50 µg/ml streptomycin at 37°C in a humidified atmosphere of 5% $CO_2$. Cells were seeded at a density of $4 \times 10^4$ cells/well and grown to approximately 60% confluency.

Inflammatory response was induced in half of the cell cultures by incubating cell monolayers with 0.39 mg/ml dexamethasone (Sigma) and 50 ng/ml human recombinant interleukin-6 (Cambridge Bioscience, UK) in DMEM supplemented with 10% FBS. After 48 hr, cells were chilled at 4°C for 30 min and then incubated with 1 mg/ml SfbI-A40-TED-iPT, or SfbI-A40-TED:Cys109Ala-iPT in PBS, or PBS for 30 min at 4°C. Cells were fixed with pre-chilled (−20°C) methanol for 20 min at −20°C, then stained with 1 mg/ml sIPD-GFP in PBS for 30 min at 25°C. Nuclei were stained with 0.5 µg/ml DAPI (15 min, 25°C). Between each incubation cell monolayers were washed extensively with sterile PBS. After 24 hr at 4°C, cells were imaged on an EVOS Digital Inverted Microscope (Life Technologies, Thermo Fisher Scientific).

## Western blot

A549 cells were grown, induced and incubated with wildtype or Cys109Ala mutant TEDs as described above. Cells were homogenized in lysis buffer (8 M urea, 5% SDS, 10% β-mercaptoethanol) by sonication in an ultrasonic water bath at 4°C for 45 min. 10 µl of cell homogenate were separated by SDS-PAGE on a 12% acrylamide gel (Bio-Rad) in Tris-glycine buffer (2.5 mM Tris, 19.2 mM glycine,

0.01% SDS, pH 8.3). The proteins were then transferred to a BioTrace PVDF membrane (PALL Gelman Laboratory) at 100 V for 1 hr in Tris-glycine buffer containing 20% (vol/vol) methanol. The membrane was subsequently washed with PBS containing 0.1% (vol/vol) Tween20 (Sigma), blocked for 1 hr with 5% (wt/vol) BSA in PBS-Tween, and incubated for 12 hr at 4°C with 1:10,000 diluted primary anti-fibrinogen α antibody (C-7, mouse IgG) (Santa Cruz Biotechnology). The membrane was washed 3 times for 10 min with PBS-Tween, and incubated with 1:20,000 diluted secondary IRDye 800CW goat anti-mouse IgG (LI-COR Biosciences), for 1 hr at room temperature. After three 10 min washes with PBS-Tween, the membrane was imaged (Odyssey CLx imaging system, LI-COR Biosciences). The same membrane was subsequently incubated for 1 hr at room temperature with 1:10,000 diluted primary anti-β-actin mouse antibody (Sigma), washed 3 times for 10 min with PBS-Tween, incubated with 1:20,000 diluted secondary IRDye 800CW goat anti-mouse IgG for 1 hr and re-imaged.

## Accession codes

Protein structures, and the data used to derive these, have been deposited at the PDB with accession numbers 5a0l (SfbI-A40-TED), 5a0n (PnTIE-TED), 5a0g (CpTIE-TED) and 5a0d (CpTIE-TED:Cys138Ala).

## Acknowledgements

This work was supported by the MRC, UK grant MR/K001485 for MW, JME, MR, MJB, USL; the BBSRC, UK grant BB/J00453 and the John Innes Foundation for MJB; the Wellcome Trust Institutional Strategic Support Fund 097831/Z/11/B for AMD; Wellcome Trust/JIF award 063597 and Wellcome Trust grants WT079272AIA and 094476/Z/10/Z to CHB for the BSRC Mass Spectrometry and Proteomics Facility; University of St Andrews and School of Biology for SYK; The Carnegie Trust for OKM. We acknowledge the Diamond Light Source for access to Data Collection facilities. We thank Eve Blumson, Aistė Skorupskaitė, Ina Schleicher and Richard Hughes for assisting with experiments, and Lee Sherry and Dan Young for the provision of A549 cells and help with cell culture. We thank Michael Gilmore for *S. aureus* VRS11b DNA; Nigel Minton for *P. difficile* CD630 DNA and Mark van der Linden and Susanne Talay for *S. pneumoniae* ST14 DNA.

## Additional information

### Funding

| Funder | Grant reference | Author |
| --- | --- | --- |
| Medical Research Council (MRC) | MR/K001485 | Miriam Walden, John M Edwards, Manfred Rohde, Mark J Banfield, Ulrich Schwarz-Linek |
| Biotechnology and Biological Sciences Research Council (BBSRC) | BB/J00453 | Mark J Banfield |
| Wellcome Trust | Institutional Strategic Support Fund 097831/Z/11/B | Aleksandra M Dziewulska |
| Wellcome Trust | JIF award 063597 | Sally L Shirran, Catherine H Botting |
| Carnegie Trust for the Universities of Scotland | Caledonian Scholarship | Ona K Miller |
| University of St Andrews | School of Biology | Su-Yin Kan |
| John Innes Foundation (JIF) | | Mark J Banfield |
| Wellcome Trust | WT079272AIA | Catherine H Botting |
| Wellcome Trust | 094476/Z/10/Z | Catherine H Botting |

The funders had no role in study design, data collection and interpretation, or the decision to submit the work for publication.

## Author contributions

MW, JME, AMD, RB, OKM, Acquisition of data, Analysis and interpretation of data, Drafting or revising the article; GS, S-YK, MW, RJJ, SLS, CHB, Acquisition of data, Analysis and interpretation of data; GJF, Conception and design, Drafting or revising the article, Contributed unpublished essential data or reagents; MR, Conception and design, Acquisition of data, Analysis and interpretation of data; MJB, US-L, Conception and design, Analysis and interpretation of data, Drafting or revising the article

# Additional files

## Supplementary files

• Supplementary file 1. Excel spreadsheet. Identification of TED-fibrinogen cross-links by MS. Each page presents data for a single TED-fibrinogen or plasma reaction. The sequences and theoretical masses of the Gln-containing TED peptide and Lys100-containing fibrinogen peptide are highlighted in red and blue respectively. The theoretical mass of the TED peptide with and without the thioester bond is given. The mass of the fibrinogen peptide is given for both the oxidized and non-oxidized (Met) states as precursor and fragment ions for both are present in the spectra. Table, top left—calculated theoretical masses for the multiply charged precursor ions of the cross-linked peptide. Precursor masses observed in the spectra are highlighted in red (*Figure 5*). Table, top right—top three hits found in the *Homo sapiens* database search for each reaction. Main tables—calculated theoretical masses of every possible fragmentation of the cross-linked peptide (orange shade). Fragment masses observed in the spectra are highlighted in red (*Figure 5*). Sequences of the fragmented peptide b- and y-ions are shown beside their corresponding masses.

• Supplementary file 2. pdf file (table). Oligonucleotide primers used in constructing expression vectors and mutants.

## Major datasets

The following datasets were generated:

| Author(s) | Year | Dataset title | Dataset ID and/or URL | Database, license, and accessibility information |
|---|---|---|---|---|
| Walden M, Edwards JM, Dziewulska AM, Kan S-Y, Schwarz-Linek U, Banfield MJ | 2015 | N-terminal thioester domain of a surface protein from Clostridium perfringens, Cys138Ala mutant | http://www.rcsb.org/pdb/search/structidSearch.do?structureId=5a0d | Publicly available at the RCSB Protein Data Bank (5a0d). |
| Walden M, Edwards JM, Dziewulska AM, Kan S-Y, Schwarz-Linek U, Banfield MJ | 2015 | N-terminal thioester domain of a surface protein from Clostridium perfringens | http://www.rcsb.org/pdb/search/structidSearch.do?structureId=5a0g | Publicly available at the RCSB Protein Data Bank (5a0g). |
| Walden M, Edwards JM, Dziewulska AM, Kan S-Y, Schwarz-Linek U, Banfield MJ | 2015 | N-terminal thioester domain of fibronectin-binding protein SfbI from Streptococcus pyogenes | http://www.rcsb.org/pdb/search/structidSearch.do?structureId=5a0l | Publicly available at the RCSB Protein Data Bank (5a0l). |
| Walden M, Edwards JM, Dziewulska AM, Kan S-Y, Schwarz-Linek U, Banfield MJ | 2015 | N-terminal thioester domain of protein F2 like fibronectin-binding protein from Streptococcus pneumoniae | http://www.rcsb.org/pdb/search/structidSearch.do?structureId=5a0n | Publicly available at the RCSB Protein Data Bank (5a0n). |

The following previously published datasets were used:

| Author(s) | Year | Dataset title | Dataset ID and/or URL | Database, license, and accessibility information |
|---|---|---|---|---|
| Pointon JA, Smith WD, Saalbach G, Crow A, Kehoe MA, Banfield MJ | 2010 | Pilus-presented adhesin, Spy0125 (Cpa), P1 form | http://www.rcsb.org/pdb/explore/explore.do?structureId=2xi9 | Publicly available at the RCSB Protein Data Bank (2xi9). |

| Author(s) | Year | Dataset title | Dataset ID and/or URL | Database, license, and accessibility information |
|---|---|---|---|---|
| Linke-Winnebeck C, Paterson NG, Young PG, Middleditch MJ, Greenwood DR, Witte G, Baker EN | 2014 | The N-terminal domain of the Streptococcus pyogenes pills tip adhesin Cpa | http://www.rcsb.org/pdb/explore/explore.do?structureId=4c0z | Publicly available at the RCSB Protein Data Bank (4c0z). |
| Kollman JM, Pandi L, Sawaya MR, Riley M, Doolittle RF | 2009 | Crystal structure of Human fibrinogen | http://www.rcsb.org/pdb/explore/explore.do?structureId=3ghg | Publicly available at the RCSB Protein Data Bank (3ghg). |

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
