## [Decision Letter]

Thank you for sending your work entitled “An internal thioester in a pathogen surface protein mediates covalent host binding” for consideration at *eLife*. Your article has been favorably evaluated by John Kuriyan (Senior editor) and two reviewers, one of whom is a member of our Board of Reviewing Editors.

The editors and the other reviewer discussed their comments before we reached this decision, and the Reviewing editor has assembled the following comments to help you prepare a revised submission.

In this study, Walden et al. defined a Cys-Gln-linked thioester-containing domain (TED) present in a large number of Gram-positive bacterial adhesin/surface proteins through bioinformatics analyses. They experimentally confirmed the internal thioester bond in 12 TEDs recombinantly produced in bacteria. They also determined the crystal structures of three TEDs (Sfbl-A40-TED, CpTIE-TED and PnTIE-TED), which are highly similar to the previously determined Cpa-TED2 structure despite the overall limited sequence similarity. This substantiates the definition of the large TED family. The second part of the story shows that SfbI-TEDs and FbaB-TED both from *S. pyogenes* can crosslink with host fibrinogen both in vitro and in the plasma, which is dependent upon the critical glutamine in the TEDs. Mass spec analyses show that Lysine-100 in fibrinogen Aα subunit reacts with the Cys-Gln thioester and forms an isopeptide bond with the glutamine side chain. Engineered *L. lactis* expressing SfbI-TED exhibit intimate adherence to fibrin and this requires the thioester Cys that activates the glutamine for being attacked by the lysine in fibrinogen. Lastly, the authors show a thioester-dependent binding of SfbI-TED to human A549 lung epithelial cells, which occurs when the cells have been stimulated with dexamethasone and IL-6 to mimic the inflammation.

The first part of the study is an expansion of previous studies (Pointon JA et al., JBC 2010 and Linke-Winnebeck C, JBC 2014), but is more thorough and compelling. The second part is significant; it is the first time to demonstrate that the thioester domain can form a covalent link with the host and to identify fibrinogen could be a host target of the TED. The only previously-known example of such a mechanism is in the human complement system. The current story represents a beautiful example of convergent evolution in which the same weapon has arisen in both bacterial pathogens and their animal hosts. Overall, the study is well designed and the data are convincing and of highly quality. The insights provided shall be interesting and significant to the field of bacteria-host interaction, and also of high value to understanding clinically important Gram-positive pathogens such as *S. pyogenes* and *Staphylococcus aureus.*

Major issues:

The reviewers have identified some weaknesses in the present study that are presented in detail below. The paper would be strengthened if these points could be addressed satisfactorily, although we shall not insist on extensive new experimentation in order to respond to these points. Please consider how you may best address these issues in a revised manuscript.

1) The paper would be improved if the authors could provide more evidence that fibrinogen is the relevant host target of one TED-containing protein during in vivo bacterial infection. Otherwise, the significance of the story is only limited to demonstrating that TED does have the ability to crosslink with another protein and the current presentation about fibrinogen could be misleading to the pathogenesis field.

2) More work is needed for substantiating the target specificity of SfbI-TEDs and FbaB-TED. In particular, the reviewers wonder if the authors have evidence that fibrinogen is the only protein in the extracellular and plasma membrane fractions that can be covalently pulled-down by SfbI-TEDs or FbaB-TED. As for the site specificity, the current mass spec data are not sufficient to demonstrate that Lysine-100 is a specific targeting site. Have the authors made further efforts to investigate the site specificity? In fact, if only Lysine-100, among numerous lysines in fibrinogen, can react with SfbI-TEDs or FbaB-TED, it will add significantly to the confidence that fibrinogen is a specific host target of the TEDs.

3) Figure 7, SfbI-A40-TED:GFP bound to A549 cells only after the cells had been pre-incubated with dexamethasone and IL-6. We understand that the treatment is designed to mimic inflammation conditions. But, why is that? Does the stimulation increase the expression and secretion of fibrinogen or provide a more favorable chemical environment for isopeptide bond formation between fibrinogen and the TED? Without further explanation, the data are quite puzzling.

Other issues to consider:

1) In the second paragraph of the subsection headed “Identification of diverse, putative thioester-containing proteins in Gram-positive bacteria”, the statement: “Both motifs are predicted to reside in a β-sheet secondary structure context”. In Figure 1—figure supplement 1, the second structure of Cpa-TED is shown for all TEDs. The authors should comment whether all the TEDs have a similar predicted second structure profile, as PSI-BLAST search is based on the conserved motif but not the second structure.

2) In the same subsection, when you state that “mutagenesis of a Cpa-TED demonstrated that neither the Gln nor Trp […] by lack of full conservation of any TQxxΦWΦxζ motif residue”, what is the function of the “TQxxΦWΦxζ α-helical motif”? It is highly conserved among the TEDs (Figure 1—figure supplement 1). While it is not essential for thioester formation in Cpa-TED, is it also the case for other TEDs? Since the authors have determined and analyzed 4 TED structures, they should clarify more about the role of this motif. Related to this, at the end of the subsection headed “Experimental validation of putative TEDs”, what do the authors mean by “this analysis lends confidence to our definition of class-II TEDs, which often lack an obvious ΦQζΦΦ motif”?

3) Thioester formation data for SaTie and CdTEP are not sufficient. We presume that the two proteins are difficult to purify and therefore the authors cannot show their total mass as they did with the other 10 TEDs. However, the methylamine modification of the glutamine does not necessarily mean that the glutamine had a thioester bond with the cysteine. Is the glutamine the only one modified by methylamine from the mass spec data? Is there any other way to show the thioester in SaTie and CdTEP or any other Class II TED (otherwise, there will only be one Class II TED experimentally confirmed in the study)?

4) In the subsection headed “TED crystal structures reveal a conserved fold and adaptations that may target different receptors”, the statement: “this may suggest a role for this loop in defining thioester target specificity.” Have the authors experimentally tested this? If so, it shall be interesting to include the data. Otherwise, it is probably a little overstated.

5) In the first paragraph of the subsection headed “Thioester-dependent adduct formation of SfbI and FbaB with the Aα subunit of fibrinogen”, the sentence: “a group of protein bands corresponding to the heterogeneous fibrinogen Aα subunit (36) were clearly depleted in the SfbI-A20-TED sample”. Why were they were not depleted in other TED reactions as their crosslinking efficiency appears to be comparable (Figure 3)? The authors should address this.

6) Figure 5—figure supplement 1 can be incorporated into the main article.

---

## [Author Response]

*1) The paper would be improved if the authors could provide more evidence that fibrinogen is the relevant host target of one TED-containing protein during* in vivo *bacterial infection. Otherwise, the significance of the story is only limited to demonstrating that TED does have the ability to crosslink with another protein and the current presentation about fibrinogen could be misleading to the pathogenesis field*.

We agree that it would be desirable to obtain data that show significance of fibrinogen binding to TEDs in infections. However, this would require a substantial experimental effort, and would be very difficult to prove unless fibrinogen knockout animal models for *S. pyogenes* nasopharyngeal colonization and invasive infections are established. This is clearly beyond the scope of our study.

We feel the significance of our study is not limited to the finding of covalent bond formation between a TED and another protein. We do demonstrate covalent bond formation between two out of twelve TEDs to one specific site in one specific protein (fibrinogen). As acknowledged by the reviewers this is the most important finding of our study. However, we report other data that we consider to be of high significance: the first experimental evidence for internal thioester bonds in highly diverse and very widespread bacterial surface proteins other than streptococcal Cpa, and the first demonstration of a role for TEDs in binding to host tissue (blood/fibrin) and host cells.

We have changed the manuscript to dispel any concern that the presentation of our data on fibrinogen binding is misleading. Binding to fibrinogen is highly specific in two respects. Firstly, there is no evidence for covalent bond formation between SfbI/FbaB-TEDs and other proteins in pull-downs from a highly complex biological sample (blood plasma). Secondly, the covalent TED binding site on fibrinogen is also highly specific. We have added new experimental evidence to further support that binding occurs through a covalent bond between TEDs and exclusively Lys100 on the Aα subunit of fibrinogen.

While we do not know at which stage of an infection streptococci would target fibrinogen through SfbI/FbaB-TED, we make two reasonable assumptions, and test these experimentally. Both the blood stream and epithelial cells are implicated in infections caused by *S. pyogenes*. The plasma pull-down and the bacteria binding experiments are of relevance for invasive infections, where bacteria would be exposed to high concentrations of fibrinogen and fibrin in the blood stream and on the surface of epithelial cells. Perhaps more interestingly, we also provide compelling evidence for a role of fibrinogen in attachment of SfbI-TED to epithelial cells. We have included additional data demonstrating the formation of a fibrinogen/TED covalent complex upon cell binding. We have rewritten the section on cell binding to clarify why we believe this is a finding of high significance for the field of bacterial pathogenicity research.

*2) More work is needed for substantiating the target specificity of SfbI-TEDs and FbaB-TED. In particular, the reviewers wonder if the authors have evidence that fibrinogen is the only protein in the extracellular and plasma membrane fractions that can be covalently pulled-down by SfbI-TEDs or FbaB-TED. As for the site specificity, the current mass spec data are not sufficient to demonstrate that Lysine-100 is a specific targeting site. Have the authors made further efforts to investigate the site specificity? In fact, if only Lysine-100, among numerous lysines in fibrinogen, can react with SfbI-TEDs or FbaB-TED, it will add significantly to the confidence that fibrinogen is a specific host target of the TEDs*.

The question of target specificity was not addressed clearly enough in the original manuscript. We have strengthened the relevant section of the paper and have added two new sets of data to support specificity. Firstly, we provide direct evidence for thioester-dependent TED-fibrinogen cross-link formation on cell surfaces (Western blot, Figure 7). Secondly, we provide additional MS data further supporting highly specific covalent bond formation between TEDs and Lys100 in the Aα subunit of fibrinogen. Lys100 is the only residue in the Aα subunit that that can be acetylated in the free form of the protein, but not when fibrinogen is bound to SfbI-TED (new Figure 5—figure supplement 1).

*3)*
Figure 7*, SfbI-A40-TED:GFP bound to A549 cells only after the cells had been pre-incubated with dexamethasone and IL-6. We understand that the treatment is designed to mimic inflammation conditions. But, why is that? Does the stimulation increase the expression and secretion of fibrinogen or provide a more favorable chemical environment for isopeptide bond formation between fibrinogen and the TED? Without further explanation, the data are quite puzzling*.

We have expanded the relevant section of our paper to address these points. Indeed, upregulation of fibrinogen expression, and its deposition on epithelial cell surfaces after treatment with dexamethasone and interleukin-6, has been established in studies cited in our manuscript. We have added new data (Western blot, Figure 7) demonstrating upregulation of fibrinogen after eliciting an inflammatory response in our experiments, in agreement with published data. The mechanism for in vitro induction of an inflammatory response in epithelial cells by the combination of dexamethasone and interleukin-6 has been described in publications cited here. We see no reason why induction of an inflammatory response in epithelial cells should change the chemical environment to favor isopeptide bond formation.

*Other issues to consider*:

*1) In the second paragraph of the subsection headed “Identification of diverse, putative thioester-containing proteins in Gram-positive bacteria”, the statement:* “*Both motifs are predicted to reside in a β-sheet secondary structure context*”*. In*
Figure 1—figure supplement 1*, the second structure of Cpa-TED is shown for all TEDs. The authors should comment whether all the TEDs have a similar predicted second structure profile, as PSI-BLAST search is based on the conserved motif but not the second structure*.

Secondary structure predictions for all TEDs, and available structural data, were used to guide the alignment (as described in the Materials and methods). We can confirm that all TEDs display similar secondary structure predictions (apart from the indels that define the two TED classes). We have added a sentence to the text making this point clearer. [Note: as stated in the figure legend, the secondary structure presented belongs to SfbI-A40-TED not Cpa-TED]

*2) In the same subsection, when you state that “mutagenesis of a Cpa-TED demonstrated that neither the Gln nor Trp […] by lack of full conservation of any TQxxΦWΦxζ motif residue”, what is the function of the* “*TQxxΦWΦxζ α-helical motif*”*? It is highly conserved among the TEDs (*Figure 1—figure supplement 1*). While it is not essential for thioester formation in Cpa-TED, is it also the case for other TEDs? Since the authors have determined and analyzed 4 TED structures, they should clarify more about the role of this motif. Related to this, at the end of the subsection headed “Experimental validation of putative TEDs”, what do the authors mean by “this analysis lends confidence to our definition of class-II TEDs, which often lack an obvious ΦQζΦΦ motif*”*?*

We are unsure if the reviewers have confused the two Gln containing motifs here, which are distinct.

As the reviewers state, the TQxxΦWΦxζ motif is conserved among TEDs and is present in each of the TED structures determined (we added a sentence to clarify this in the section describing the structures). While the TQxxΦWΦxζ motif is located close to the thioester bond, the structures do not allow any definitive conclusions concerning the putative role of this motif in bond formation, and we do not wish to speculate about it here. It is not our primary focus, and beyond the scope of our study. However, we are presently working on this in our labs. As for Cpa-TED, preliminary data for SfbI-A40-TED supports a non-essential, and so far unknown, role for this motif.

The ΦQζΦΦ motif contains the Gln residue that, with the Cys, forms the internal thioester bond. The lack of a well-defined ΦQζΦΦ motif in class II TEDs is apparent from the sequence alignment in Figure 1—figure supplement 1. To ensure clarity, we have added a reference to this figure at the appropriate place in the text.

*3) Thioester formation data for SaTie and CdTEP are not sufficient. We presume that the two proteins are difficult to purify and therefore the authors cannot show their total mass as they did with the other 10 TEDs. However, the methylamine modification of the glutamine does not necessarily mean that the glutamine had a thioester bond with the cysteine*. *Is the glutamine the only one modified by methylamine from the mass spec data? Is there any other way to show the thioester in SaTie and CdTEP or any other Class II TED (otherwise, there will only be one Class II TED experimentally confirmed in the study)?*

We have added new experimental intact mass data for SaTIE-TED and CdTEP-TED to Table 1, showing that these proteins are ∼17 Da lighter than their theoretical mass. These data, coupled with the specific modification of a single Gln in the proteins by methylamine, is strong evidence for formation of a thioester in these proteins. We can confirm that only a single Gln is modified by methylamine, as determined by mass spec. In absence of a structure, methylation of a Gln in a TED is, in fact, the best evidence for a thioester bond since it distinguishes this bond from other cross-links such as isopeptides, which also give rise to a –17 Da difference in mass. Amide carbonyl carbons are poor electrophiles and will not react with methylamine, unless the carbonyl group is activated, which in the case of TEDs is achieved by transformation of the amide bond into a thioester.

*4) In the subsection headed “TED crystal structures reveal a conserved fold and adaptations that may target different receptors”, the statement:* “*this may suggest a role for this loop in defining thioester target specificity.*” *Have the authors experimentally tested this? If so, it shall be interesting to include the data. Otherwise, it is probably a little overstated*.

Given that we only have only identified one TED target so far, and that the structure of a TED complex has yet to be determined, we could not test this hypothesis. We have altered the text to avoid overstatement.

*5) In the first paragraph of the subsection headed “Thioester-dependent adduct formation of SfbI and FbaB with the Aα subunit of fibrinogen”, the sentence:* “*a group of protein bands corresponding to the heterogeneous fibrinogen Aα subunit (*[36]*) were clearly depleted in the SfbI-A20-TED sample*”*. Why were they were not depleted in other TED reactions as their crosslinking efficiency appears to be comparable (*Figure 3*)? The authors should address this*.

The gels presented in Figure 3 reflect differences in TED reactivity under a standardized set of conditions (reaction times, incubation temperatures, pH, fibrinogen and TED concentrations). Under alternative assay conditions, it is possible to achieve depletion of the fibrinogen Aα band for TEDs other than SfbI-A20-TED, but this requires optimization for individual proteins. To avoid any misinterpretation or overstatement, we have changed the text to clarify this and stated that the important conclusion from this was to focus our attention on the fibrinogen Aα subunit for subsequent experiments.

*6)*
Figure 5—figure supplement 1
*can be incorporated into the main article*.

We have now included the original Figure 5—figure supplement 1 in the main Figure 5.